



# Estimating extreme river discharges in Europe through a Bayesian Network

Dominik Paprotny[1], Oswaldo Morales Nápoles[1]

[1]Department of Hydraulic Engineering, Faculty of Civil Engineering and Geosciences, Delft University of Technology, Stevinweg 1, 2628 CN Delft, The Netherlands.

*Correspondence to*: Dominik Paprotny (d.paprotny@tudelft.nl)





**Abstract.** Large-scale hydrological modelling of flood hazard requires adequate extreme discharge data. Models based on physics are applied alongside those utilizing only statistical analysis. The former requires enormous computation power, while the latter are most limited in accuracy and spatial coverage. In this paper we introduce an alternate, statistical approach based on Bayesian Networks (BN), a graphical model for dependent random variables. We use a non-parametric BN to describe the joint distribution of extreme discharges in European rivers and variables describing the geographical characteristics of their catchments. Data on annual maxima of daily discharges from more than 1800 river gauge stations were collected, together with information on terrain, land use and climate of catchments that drain to those locations. The (conditional) correlations between the variables are modelled through copulas, with the dependency structure defined in the network. The results show that using this method, mean annual maxima and return periods of discharges could be estimated with an accuracy similar to existing studies using physical models for Europe, and better than a comparable global statistical method. Performance of the model varies slightly between regions of Europe, but is consistent between different time periods, and is not affected by a split-sample validation. The BN was applied to a large domain covering all sizes of rivers in the continent, both for present and future climate, showing large variation in influence of climate change on river discharges, as well as large differences between emission scenarios. The method could be used to provide quick estimates of extreme discharges at any location for the purpose of obtaining input information for hydraulic modelling.

**Keywords.** hydrology; catchments; floods; copulas; climate change; return periods; flood risk

## 1 Introduction

There is currently substantial concern in Europe about increasing flood risk linked mainly to climate change. Available studies (Whitfield 2012, Feyen et al. 2012, Alfieri et al. 2015) predict that the severity of floods will increase, due to changes in extreme precipitation and socio-economic development. Abundant availability of continental and global climate, land use and elevation data result in many studies analysing floods in the same large domain. However, the amount of hydrological observations at disposal is far from sufficient. This is not only the result of an uneven network of measurement stations, but also the limited dissemination of data by national or local bodies responsible for their collection. At the same time, they are necessary to calculate accurate hydrological event scenarios for the purpose of delimiting flood zones. Those scenarios are typically values of extreme river discharge or water level with a certain return period, i.e. the average interval of time between the occurrences of an event with the same magnitude. Such calculation additionally require long data series, further narrowing the number of locations were such analysis can be performed. In effect, large-scale flood studies must fill the gaps in measurements with modelled river flows. There are two main groups of methods used to obtain discharge values in ungauged catchments.

The first group of methods are rainfall-runoff models. They utilize physical equations describing processes such as infiltration, runoff, retention in order to transform rainfall into river discharges. A large number of models have been used in smaller scales, though in recent years few studies applied them on a continental or global scale. One series of publications





(Dankers and Feyen 2008, Rojas et al. 2012, Alfieri et al. 2014 and others) presented calculations using LISFLOOD model. The simulation was set up for Europe with a 5-km resolution. Many different datasets of rainfall amount were tested, including historical observations and future climate simulations, deriving daily discharge data for most of the continent. Another group of studies (Ward et al. 2013, Winsemius et al. 2013) has presented a global hydrological model GLOFRIS.

This model has a much coarser resolution than the previous one—the rainfall-runoff module uses a 0.5° grid (ca. 50–80 km resolution over Europe). All those studies used the results to perform an extreme value analysis of modelled river discharges, and some also carried on with a flood hazard study. The main drawback of this approach is the computational expense, which necessitates a reduction in resolution (though it should be noted that the main driver—meteorological data—also have limited resolution). Additionally, only a limited number of rivers is included. For instance, LISFLOOD-based studies used a

threshold of 1000 km$^2$ catchment size, later reduced to 500 km$^2$. Meanwhile, GLOFRIS was prepared only for rivers with Strahler order 6 or above, which is about a third of the river length included in the aforementioned European model.

The second group of approaches are statistical methods, of which a large variety exists. Several methods are based on the fact that catchments close to each other share many characteristics. River basins are therefore pooled into groups based on geographical proximity alone or also based on catchment size, climate data, terrain or soil type. However, the studies

employing such techniques mostly covered a limited domain, typically single countries (Meigh et al. 1997, Salinas et al. 2013). The first global analysis was recently presented by Smith et al. (2015). The study applied regional frequency analysis (RFA) for all continents for the first time. Here, after clustering catchments based on size, climate type and average rainfall, a probability distribution of discharges is calculated for each region. Estimates of extreme discharges for a given ungauged catchment are derived by first assigning it to a proper region. Then using data on catchment size and rainfall together with

region-specific coefficients a simple regression equation is solved in order to obtain an estimate of the mean of annual maxima of discharges in the catchment. Lastly, a Generalized Extreme Value probability distribution with region-specific parameters is used to calculate return periods of discharges. Flood scenarios (though peak discharges only) obtained through this method were used for a global flood hazard analysis through LISFLOOD model by Sampson et al. (2015).

There are also several statistical methods that rely solely on the geographical characteristics of catchments to estimate

discharges. Many of them are simple equations that can be easily applied to quickly solve practical problems in engineering, such as estimating dike heights or calculating necessary channel or culvert capacity. Also, they are typically only applicable in small areas for which they were prepared. Usually, they are a variation of the "rational equation", which states that river discharges can be calculated by multiplying catchment area, rainfall intensity and runoff coefficient (Chow 1988, Sando 1998). The first two elements are used in virtually all methods. The remaining element is either left out due to the difficulty

of estimating it, or is derived from a model table of coefficients, or additional factors are added as proxies. For instance, Stachý and Fal (1986) developed an equation to calculate 100-year discharge in catchments above 50 km$^2$ in Poland which incorporates seven factors: catchment area, extreme rainfall (100-year return period), soil type, catchment slope, river slope, lake area and marsh area. However, it also requires an additional empirical coefficient for each physiogeographic region of the country, while different return periods than the default 100 years are obtained by multiplying discharge by a region-



specific factor, similar to the RFA method. Another example is the preliminary flood risk assessment in Norway (Peereboom et al. 2011), which utilized a simple regression between catchment area and 500-year water level. An "envelope curve" approach was then applied, in which a curve is constructed in such a manner that it contains all (or almost all) observations. This concept was long used to make crude estimations of maximum possible floods, also on continental scale (e. g. Padi et al. 2011 applied it to Africa). Finally, some attempts were made to apply multiple linear regressions, also on global scale (Herold and Mouton 2010).

This paper presents a new statistical method to calculate extreme river discharges under present and future climate in Europe. It was devised as an alternative to existing physical and statistical models; its purpose was to provide boundary conditions for hydraulic modelling that could be used in a pan-European flood hazard analysis. The method is based on Bayesian Networks that combine probability theory and graph theory in order to build and operate a joint distribution. It is used to analyse and represent the dependencies between different environmental variables, including river discharges. We present the quantification of the model based on a large dataset of river gauge observations and pan-European spatial datasets. The model shows good performance across regions of Europe at different time periods. We also present a comparison of this new approach with other methods, both physical and statistical. Lastly, we apply it over the entire domain to obtain a large database of extreme discharges, and analyse the influence of climate change on their return periods.

An early and preliminary variant of the method was originally reported in Paprotny and Morales Nápoles (2015). The Bayesian Network presented there is superseded by an improved version described herein. Also, the work is part of a bigger effort to create pan-European meteorological and hydrological hazard maps under "Risk analysis of infrastructure networks in response to extreme weather" (RAIN) project. This fact influenced the choice of the domain and input data, which is explained in the next sections, though this does not limit the applicability of the method outside the domain.

## 2 Materials and methods

This section first gives an overview of the model's framework and elements, and then proceeds with details on how the model was prepared, what datasets were used to build it, what are the underlying mathematical methods and how its accuracy and utility was assessed.

### 2.1 Workflow and outline of the method

The basic elements of the procedure to derive extreme discharge estimates through a Bayesian Network are presented in Fig. 1. The first step was identifying available data on annual maxima ($Q_{AMAX}$) of daily river discharge (I), and also catchments which contribute to locations where the measurements were made (section 2.2), i.e. gauged catchments (II). Then, several large-scale (pan-European or global) spatial datasets were compiled (III), providing information on most important variables influencing extreme river flow behaviour (section 2.3), both for gauged and ungauged catchments (IV). The dependence between those variables and river discharges, were analysed through copulas and Bayesian Networks



(section 2.4) (V). After extensive testing of different configurations, an optimal model was constructed (section 2.5) that had the highest performance in validation in terms of the underlying statistical model and prediction capability (sections 2.7 and 3.1) (VI). The output of the model are annual maxima of daily river discharges (VII), which were then fitted to a probability distribution in order to obtain return periods (section 2.6). After the method was ready, it was applied for all catchments (IV)

in the domain to create a database of discharges (VIII). Frequency analysis allowed then to obtain return periods of discharges under present and future climate in Europe (section 3.2) (IX). The accuracy of the Bayesian Network model was also contrasted with alternate methods (section 4.1) (X).

## 2.2 River discharge data

Discharge data from measurements stations were collected over a domain covering most of Europe (Fig. 2). Because

this research focuses on European Union (EU) countries, all river basins at least partially located in this group of states are included (including Cyprus, geographically part of Asia). Some additional neighbouring basins were added for complete coverage of Europe except for the territory of the former Soviet Union. Only the outermost regions of Madeira, the Azores and the Canary Islands were omitted because they are outside the EURO-CORDEX climate model; notwithstanding, their river networks are very limited and therefore of little interest.

In total, series for 1841 stations were compiled, not including a few dozens of available stations whose tributaries could not be unequivocally identified and were therefore not included in the analysis. The data were collected from five sources, as follows:

- 1186 stations from the Global Runoff Data Centre (2016);
- 82 stations from the Norwegian Water Resources and Energy Directorate (2015);
- 284 stations from the Swedish Meteorological and Hydrological Institute (2016);
- 239 stations from Centro de Estudios Hidrográficos (2012);
- 50 stations from Fal (2000).

The data collected were daily discharges observed between 1950 and 2013, though of primary interest were data up to 2005, since it was the maximum range of EURO-CORDEX climate models' historical scenario runs. All datasets were

quality-checked by the providers, though a few erroneous records had to be corrected after inspection. Daily discharges were transformed into annual maxima ($Q_{AMAX}$) for each calendar year, except for the last group of 50 stations, as Fal (2000) only reported the extreme and mean values. Total number of $Q_{AMAX}$ values for years 1950–2005 in the database was 74,757. The stations represent 37 countries and 439 different river basins (78% of the domain's area of 5.67 million km$^2$). However, the south-eastern part of Europe is substantially underrepresented, with most stations concentrated in Scandinavia and the

Western Europe. France has the highest number of $Q_{AMAX}$ values in the database (14%), followed closely by Spain, Sweden and United Kingdom, as can be seen in Table 1. The catchments' size spans from 1.4 to 807,000 km$^2$, with a biggest group of them being in the 100–1000 km$^2$ range.



Long data series, i.e. at least three full decades of uninterrupted data (1951–80, 1961–90 or 1971–2000) were available for 1125 stations. These observations were used to validate the accuracy of the model in estimating mean $Q_{AMAX}$ and return periods, while the complete database was used to quantify the Bayesian Network model.

## 2.3 Spatial datasets

Several large-scale spatial datasets were collected for this work, even though not all of them were used in the final set-up of the model. Nevertheless, all were useful for testing different configurations of the BN. The most important dataset is a map of the river network and catchments, which was derived from pan-European CCM River and Catchment Database v2.1, or CCM2 (Vogt et al. 2007, de Jager and Vogt 2010). It was created by calculating flow direction and accumulation on a 100-m resolution digital elevation model (DEM), combined with land cover information, satellite imagery and national GIS

databases. CCM2 was utilized to delimit the domain used in this paper; in total that area covers 831,125 river sections (almost 2m km in length) in 70,638 river basins. Each river gauge station was connected with a corresponding river section in CCM2. Each river section belongs to one primary catchment, whose attributes includes the identifier of the next downstream catchment. Using this information, the whole tributary of a gauge station, or any other point in the domain, could be delimitated. For each catchment, by employing GIS software, various statistics were calculated based on other

datasets described below. Additionally, a few indicators could be derived from this dataset alone: catchment area, river network density (total river length divided by catchment area) and catchment circularity (catchment area divided by the area of a circle that has the same perimeter as the catchment).

The next most relevant source of information are climate data, both historical and future projections. Two datasets for the former were analysed. E-OBS is an spatial interpolation of observations made by weather stations covering years 1950–

2015 (Haylock et al. 2008), while ERA-Interim is a complete climate reanalysis for 1979–2015 (Dee et al. 2011). However, E-OBS has gaps in spatial coverage and includes few variables, whereas ERA-Interim has a relatively coarse resolution (0.75°). In effect, slightly better performance of the model was recorded using high-resolution control runs of a climate model under EURO-CORDEX framework (Jacob et al. 2014). EURO-CORDEX uses regional climate models (RCM) for Europe, where boundary conditions are obtained from global-scale general circulation models (GCM). In this work, we

utilize simulations for the historical run (1950–2005) and two climate change scenarios (RCP 4.5 and RCP 8.5 for 2006–2100). They were made by Climate Limited-area Modelling-Community utilizing EC-Earth general circulation model (run by ICHEC) with COSMO_4.8_clm17 regional climate model (Rockel et al. 2008), realization r12i1p1. This RCM was chosen because of its relatively good performance model in estimating extreme precipitation in comparison with others (Kotlarski et al. 2014) and also due to the earlier start of historical runs than in many other models (1950). Moreover, it was

found by Rojas et al. (2012) that the influence of differences models on the results of hydrological simulations in Europe is rather small. Therefore, only this model was used in the analysis instead of a whole ensemble. Also, no bias correction was performed, even though it is often considerable for extreme precipitation (Rojas et al. 2011). For the sake of simplicity and universality of the method we opted for using all input data unaltered. From this dataset four variables were derived: total





precipitation, snowmelt, near-surface temperature and total runoff. All data were daily values on a 0.11° rotated grid (spatial resolution of about 12 km).

Meteorological factors are the driving force behind floods, but more factors influence the runoff – terrain, land use and soils. Information on terrain was obtained from two digital elevation models. Most of the domain is available from EU-

DEM, a dataset produced for the European Environment Agency. It was created by merging two sources of satellite altimetry data – Shuttle Radar Topography Mission (SRTM) and ASTER GDEM. It has a 25 m resolution and covers 39 countries (DHI GRAS 2014), including areas north of 60° N missing from SRTM-only datasets. For Eastern Europe and some Atlantic islands which are not covered by EU-DEM, SRTM data where used instead (Farr et al. 2007). This model has a 3 arc second resolution (~100 meters over Europe) and has several versions available. The one used here is a void-filled derivate obtained

from Viewfinder Panoramas (2014). Both datasets were resampled to a common 100 m grid matching the CCM2 dataset. The variables calculated from the DEMs included average elevation, average river slope and average catchment slope. The latter was derived in the ways: by averaging all slopes in DEM, or calculating the slope $S$ with the equation:

$$S = \frac{H_{max} - H_{min}}{\sqrt{A}} \tag{1}$$

where $H_{max}$ is the maximum, and $H_{min}$ the minimum, elevation in the catchment and $A$ is the catchment area. Another

variable, the time of concentration, which is a measure of water circulation speed in the catchment, was calculated based on Gericke and Smithers (2014). Finally, we tested terrain classification similar to one used in FLEX-Topo hydrological model (Savenije 2010). In this approach, all grid cells in the DEM are classified based on height-above-nearest-drainage (HAND), slope inclination and absolute elevation (Gharani et al. 2011, Gao et al. 2014). Three classes—wetlands, hillslopes and mountains—were calculated as percentage of total catchment area.

Land use statistics for catchments were mainly based on CORINE Land Cover (CLC), another dataset produced by the European Environment Agency (2014a). CLC 2000 edition, version 17 (12/2013) in raster format (100 m resolution) was used here. It includes 44 land cover classes with a minimum mapping unit of 25 ha and covers 39 countries. The main source material were Landsat 7 satellite images from years 1999–2001 (European Environment Agency 2007). Similarly to EU-DEM, the dataset does not cover some catchments in Eastern Europe and few other areas. Missing information was

supplemented by Global Land Cover 2000 dataset, produced by the Joint Research Centre using algorithmic processing of SPOT 4 satellite images (Bartalev et al. 2003). This product has a 30 arc second resolution and includes 22 land cover classes. The different classifications were synchronised to derive the area covered by forests, croplands (total and irrigated), marshes, lakes, glaciers, bare land and artificial surfaces. However, the data is only available for a single year for the whole domain, even though CLC was produced also for 2006 and, in some countries, for 1990. In contrast to terrain or soils, land

use is dynamic and could influence the analysis for early time periods. Some historical land-use reconstructions and projections (e. g. Klein Goldewijk et al. 2011) do not have the necessary resolution or thematic coverage for use in this analysis. Therefore, a fixed values of land use percentages was used for all years, including climate change scenarios.





Last but not least, data on soil properties were analysed. Occurrence of peat, unconsolidated and Eolian deposits, average water content and soil texture were derived from European Soil Database v2.0 (Panagos et al. 2012), developed at 1:1,000,000 scale, and Harmonized World Soil Database v1.2 (FAO/IIASA/ISRIC/ISS-CAS/JRC 2012), available at 30 arc second resolution. Also, soil sealing (i.e. area covered by artificial impervious surfaces) was obtained from Revised Soil

Sealing 2006, a dataset based on satellite imagery with a 100 m resolution (European Environment Agency 2014b). Additionally, grain-size structure of the soil (gravel, sand, silt, clay) was calculated from SoilGrids1km database (Hengl et al. 2014).

## 2.4 Bayesian Networks

As noted in the introduction, Bayesian Networks (BN) are graphical, probabilistic models (Pearl 1988, Kurowicka and

Cooke 2006). They have several advantages, compared to other methods, for the application described in this paper. For one thing, its graphical nature makes the dependence configuration explicit, as evidenced in Fig. 3 in the next section. It captures, for example, dependencies between different environmental variables, which are not easily modelled with regression methods. Also, it allows to capture the often non-linear nature of those dependencies. The class of BNs used in this research includes several elements, whose specifics need to explained before the actual hydrological model is presented.

First of all, consider a set of random variables $(X_1, X_2, \ldots, X_n)$, which could be discrete, continuous, or both. This distinction defines the different types of BNs. Here, more suitable is a continuous BN, since our environmental data are of this sort. Also, discrete BNs are only efficient for small models with variables having a limited number of states. That is because of the way the (conditional) probabilities are calculated, as we explain later on. The random variables are represented as "nodes" of the BN, while the dependencies between them are represented as "arcs" joining different nodes.

An arc represents the (conditional) correlation between two variables, and has a defined direction. The node whose arc points into the direction of another node is known as the "parent", while the node on the other end of the arc is its "child". A set of notes and arcs forms the eponymous "network" of the BN. The arcs have to connect the nodes in such a manner that the graph is acyclic, i.e. if we chose any node and follow strictly the direction of all arcs in a path, we will not end up in the same node. Each variable is conditionally independent of all its predecessors given its parents. Therefore, each variable has a

conditional probability function given its parents, and the joint probability can be expressed as:

$$f(X_1, X_2, \ldots, X_n) = \prod_{i=1}^{n} f(X_i | Pa(X_i)) \tag{2}$$

where is $Pa(X_i)$ is the set of parent nodes of $X_i$, with $i = 1, \ldots, n$. Naturally, if there are no parents, $f(X_i | X_{pa(i)}) = f(X_i)$. We already see that one of the purposes of BNs, perhaps the main one, is updating the probability distributions of subsets of nodes, when evidence (observations) of a different subset becomes available. Hence, it is important not only to

properly set-up the network with nodes and arcs, but also choosing an optimal method to describe the dependencies. In case of a discrete network, this is done using conditional probability tables. Let's assume we use 8 nodes, of which one has 7




parents, which is the actually case with the BN we describe in the next section. Now, we discretize all nodes into only 5 states, which in case of e. g. river discharges means assigning the values into classes a whole order of magnitude apart. In order to quantify all dependencies for this one particular node, we need $5^8 = 390,625$ conditional probabilities—several times more than the available number of samples.

5        Meanwhile, using a non-parametric continuous BN, we only need to specify an empirical marginal distribution for each variable and a rank correlation for each arc (Hanea et al. 2015). Spearman's rank correlations are used to parameterize one-parameter (conditional) copulas. A copula is, loosely, a joint distribution on the unit hypercube with uniform (0,1) margins. There are many types of copulas, described in detail by Joe (2014). Here, we use bivariate Gaussian copulas, an assumption that was tested against alternate distributions. Details of this calculation, and the validation of the whole Bayesian Network

can be found in the Supplement. The bivariate Gaussian copula $C$ has the following cumulative distribution function:

$$C_\rho(u,v) = \Phi_\rho\big(\Phi^{-1}(u), \Phi^{-1}(v)\big), (u,v) \in [0,1]^2 \qquad (3)$$

where $\Phi$ is the standard or multivariate Gaussian cumulative distribution and $\rho$ is the (conditional) product moment correlation between the two marginal probability distributions $u$ and $v$ in the interval [0,1]. In contrast, the non-parametric BN we apply here is parametrized by (conditional) rank correlations. This is because they are algebraically independent;

hence, any number in the interval [-1,1] assigned to the arcs of the BN will warranty a positive definite correlation matrix. The rank correlation (denoted by $r$ ) of two random variables $X_i$ and $X_j$ with cumulative distribution functions $F_{X_i}$ and $F_{X_j}$ is the usual Pearson's product moment correlation $\rho$ computed with the ranks of $X_i$ and $X_j$, i.e.

$$r\big(X_i, X_j\big) = \rho\left(F_{X_i}(X_i), F_{X_j}(X_j)\right) \qquad (4)$$

For a one-parameter bivariate copula, eq. 4 becomes:

20                $$r\big(X_i, X_j\big) = 12 \int_0^1 \int_0^1 C_\theta(u,v)\, du\, dv - 3 \qquad (5)$$

The conditional rank correlation of $X_i$ and $X_j$ given the random vector $\mathbf{Z} = \mathbf{z}$ is the rank correlation calculated in the conditional distribution of $(X_i, X_j|\mathbf{Z} = \mathbf{z})$. For each variable $X_i$ with $m$ parents $Pa_1(X_i), \dots, Pa_m(X_i)$ the arc $Pa_j(X_i) \rightarrow X_i$ is associated with the rank correlation:

$$\begin{cases} r\left(X_i, Pa_j(X_i)\right), & j = 1 \\ r\left(X_i, Pa_j(X_i)|Pa_1(X_i), \dots, Pa_{j-1}(X_i)\right), & j = 2, \dots, m \end{cases} \qquad (6)$$

where the index $j$ is in the non-unique sampling order. For more details on the non-parametric Bayeian Networks we refer the reader to Hanea et al. (2015). Having all the variables and parameters of the Bayesian Network in place the joint distribution is uniquely determined. Under the Gaussian copula assumption, exact inference is available as well as efficient sampling procedures (for details, see Hanea et al. 2006). Here, 1000 samples were used each time we wanted to conditionalize the BN in order to derive an estimate of river discharges for a given location in our dataset. This number of

samples is adequate to approximate the conditional distributions of interest while keeping the procedure computationally



feasible. The Bayesian Network for river discharges presented here was implemented in Matlab software; UniNet programme for non-parametric BNs was also used to visualize and analyse the model (for details, see Morales Nápoles et al. 2013).

## 2.5 Extreme discharge model

The final BN for extreme river discharge was derived through testing many configurations involving around 30 variables. The BN cannot be created in an automated manner, nor is it desirable to do so. Therefore, it was build stepwise and assessed using a set of statistical measures presented in section 2.7. The final model uses 8 variables and is presented in Fig. 3, with a histogram representing each variables' distributions. The position of the nodes shows their hierarchy relative to the annual maximum of daily river discharge (*MaxDischarge*): the order in which different variables conditionalize on the

river discharge distribution (using eq. 6) is clockwise. The (conditional) rank correlation coefficients are indicated at the arcs. The variables are described below.

    **Annual maximum of daily river discharge** (*MaxDischarge*) in m³/s. The parents of this variable are all the remaining variables in the BN. By far the most important is the **catchment area** (*Area*) in km$^2$. It determines the scale of the processes in a river basin and is largely dependent on **catchment steepness** (*Steepness*) in m/km. This is because mountainous

catchment are very small, divided by ranges, and only grow in size when many rivers join along the way to its drainage basin, crossing more planar regions. Steepness was calculated here using eq. (1); it is a proxy for terrain characteristics that influences the speed with which the water from rainfall moves down the slopes (Savenije 2010).

    The climate model from EURO-CORDEX framework delivered two variables to the BN. First is the **annual maximum of daily precipitation and snowmelt** (*MaxEvent*) in mm. Both factors are relevant, though melting of snow cover is

important only regionally. Both events also often occur concurrently (as evidenced in the list of European floods by Barredo 2007), so using a summation of the two improved the performance of the BN. The variable has one parent, catchment steepness, as hilly and mountainous areas receive more precipitation, also in form of snow. The second variable is the **extreme runoff coefficient** (*RunoffCoef*), a dimensionless indicator. It was constructed to include meteorological factors influencing the circulation of water in a catchment. Every climate model needs to represent this variable, taking into account

factors such as soil moisture, evaporation and retention. The annual maximum of climate model variable "total runoff" was obtained for each sample, and then divided by *MaxEvent*. This variable is dependent on catchment steepness, since in hilly/mountainous terrain conditions limit evaporation or retention. It should be noted that values these climate variables were calculated as an average of annual maxima derived for each grid cell separately, and not by identifying the largest single event that occurred in the catchment.

The BN is completed by three land cover types, all expressed as % of total catchment area. The statistics were obtained by choosing relevant classes from land cover datasets. First variable are **lakes**, which were obtained using "water bodies" class in Corine Land Cover (CLC), with missing coverage supplemented by the water body layer in CCM2 database. Lakes retain water from rainfall or snowmelt, thus reducing river discharge. This node has two parents, catchment steepness and




extreme runoff coefficient. Lakes, especially large, are more prevalent in post-glacial plains of northern Europe, though increase lake cover is observed also in the mountains. In both those areas runoff coefficient is higher, with the same factors influencing both (like soils or temperature). Second variable are **marshes**, which are defined by CLC as three classes "inland marshes", "peat bogs" and "salt marshes", while from Global Land Cover 2000 (GLC) "regularly flooded shrub and/or

herbaceous cover" class was used here. Similarly to lakes, marshes increase retention in a catchment. They often occur in the same areas as lakes, with soils and climatology also having influence (as estimated by the runoff coefficient). Lastly, the **build-up areas** (*Buildup*) variables contain the "artificial surfaces" class from CLC or GLC. Construction seals the soil, reducing infiltration, while water management systems collect the rainfall and routes it directly to river. This variable is influence, in order, by catchment steepness (flat areas are preferred for construction), runoff coefficient (which is higher is

colder areas), lakes and marshes (less space available for construction).

      In order to estimate river discharge in an ungauged catchment, then BN is updated, i.e. the value a node or set of nodes is defined based new evidence . Fig. 4 shows the effects of updating on the example of Basel station in Switzerland (meteorological data pertain to year 2005). Conditionalizing on only two variables: catchment area and steepness changed the mean of the distribution from 341 to 1740 m³/s. Knowing all seven variables that are parents of the river discharge node,

we obtain an estimate of river discharge of 2819 m³/s. In this case, the estimate is fairly accurate, as discharge of 3212 m³/s was actually measured. The same procedure was applied to all rivers in the domain. Additional examples of conditionalization of the BN can be found in the Supplement. It should be noted that the discharge in each river section was estimated independently from another section in the same river using data for the entire upstream area.

**2.6 Return periods of discharges**

      Annual maxima of daily river discharges calculated using the BN allowed to performed a frequency analysis. For validation of the model with observations only stations with long data series were used. 30-year time periods were used in the calculation, and so the main validation set consists of 958 stations with 1971–2000 data, 129 with 1961–90 data and 38 with 1951–80 data. All stations with data for a given time period, and subsets comprising different regions of Europe and

catchment size were also analysed. To find an optimal model for estimating the marginal probability distribution of annual maxima of discharges, we used Akaike Information Criterion measure (Mutua 1994). It showed significant variability among stations. On average, Generalized Extreme Value (GEV) distribution was the best fit, though by only a small margin over several other distribution. This three-parameter distribution, however, gave very large errors for some stations. Therefore, to avoid completely unrealistic estimates in the database, we decided to use the two-parameter Gumbel distribution, which is

basically the GEV distribution with the shape parameter equal zero. This distribution was used in several large-scale flood hazard studies (Dankers and Feyen 2008, Hirabayashi et al. 2013, Winsemius et al. 2013, Alfieri et al. 2014). In order to calculate discharge $Q$ with probability of occurrence $p$, the following equation is used:





$$Q_p = \mu - \sigma \ln(-\ln(1 - p)) \tag{7}$$

where $\mu$ is the location parameter and $\sigma$ is the scale parameter. Parameters were fitted using maximum likelihood estimation (Katz et al. 2002).

**2.7 Measures for validation of the model's results**

5      Accurate estimation of return periods of extreme discharges, as well as mean annual maximum, are the desired outcomes of the Bayesian Network model. Quality of return period and average maxima simulations was evaluated using a set of three measures: coefficient of determination, Nash-Sutcliff efficiency and RMSE-observations standard deviation ratio. Those methods were selected because they were also used in other studies (e.g. Rojas et al. 2011) and they were included in an overview of most important measures by Moriasi et al. (2007). Firstly, the Pearson's coefficient of

10     determination ($R^2$) was used to measure the correlation between observed and simulated values. In Kurowicka and Cooke (2006) it is noted that $R^2$ actually factorizes into a function of the conditional rank correlations attached to the BN. Secondly, Nash-Sutcliffe efficiency ($I_{NSE}$) was applied to measure bias of the model. Its maximum value is 1, which means a plot of observed vs simulated data fits the 1:1 line (no bias), while a value below 0 (down to $-\infty$) indicates that the mean of the observations is a better predictor than the simulated value. The relevant equation is as follows:

$$I_{NSE} = 1 - \left[ \frac{\sum_{i=1}^{n} \left( x_i^{obs} - x_i^{sim} \right)^2}{\sum_{i=1}^{n} (x_i^{obs} - x^{mean})^2} \right] \tag{8}$$

where $x_i^{obs}$ is the $i$-th observation of a variable, $x_i^{sim}$ is the $i$-th simulated value of that variable and $x^{mean}$ is the mean of observations. The final measure is root mean square error ($I_{RMSE}$)-observations standard deviation ratio ($I_{RSR}$). It standardizes the RMSE based on the standard deviation of observations ($I_{SDobs}$):

$$I_{RSR} = \frac{I_{RMSE}}{I_{SDobs}} = \frac{\sqrt{\sum_{i=1}^{n} \left( x_i^{obs} - x_i^{sim} \right)^2}}{\sqrt{\sum_{i=1}^{n} (x_i^{obs} - x^{mean})^2}} \tag{9}$$

20   **3 Results**

In this section, extreme river discharges calculated using the Bayesian Network are compared with observations. Additionally, we present the results of applying the method to estimate the influence of climate change on discharges in Europe.



## 3.1 Validation of the model's results

Extreme river discharges estimates obtained from the Bayesian Network are presented and compared with observations in Fig. 5. The graphs include the mean annual maximum of daily discharge ($Q_{MAMX}$) and three return periods of discharges. The former shows the highest performance with both $R^2$ and $I_{NSE}$ at 0.92, while accuracy of simulated discharge fitted to

Gumbel distribution decreases with the probability of occurrence. 2-year discharge has the same performance as $Q_{MAMX}$, while the 1000-year discharge is noticeably becoming biased, mainly for very large rivers. It should be also remembered that the return periods were based only on 30-year series, therefore a 100- or 1000-year discharge includes the uncertainty of extrapolation of the return periods. However, the $I_{NSE}$ value is still very good, and $R^2$ changes relatively little.

Performance of the model was analysed also in more detail, by time period, region or catchment area (Table 2). For four

different time periods, where availability of stations varies, the results of the validation are almost identical. Only for 1981–2010 it is slightly lower, because it is partially outside the timespan of the historical scenario of EURO-CORDEX; data from RCP 4.5 climate change scenario run had to be used to fill the missing information for 2006–2010. Much more variations in the quality of simulations is observed when dividing the results by geographical region (their definitions correspond to the regionalisation of the CCM2 catchment database). Western Europe (comprised mainly of France, Belgium, the Netherlands and Rhine river basin) had particularly good results for $Q_{MAMX}$, followed by Danube river basin and Scandinavia (roughly

defined as Sweden and Norway). The lowest correlation for $Q_{MAMX}$ was observed in the Iberian Peninsula (Spain and Portugal), while Central Europe (mainly Poland, Lithuania, Denmark and north-east Germany) had the lowest bias. Iberia had the lowest performance for 100-year discharge, while Western Europe recorded the highest correlation, and Scandinavia the best score in $I_{NSE}$ and $I_{RSR}$. Central European and Scandinavian stations' bias and error was lower for 100-year return

period compared to $Q_{MAMX}$. No region dropped below acceptable levels, albeit stations in the Iberia and "other regions" have noticeably lower performance. In case of Spain, to which almost all stations collected for the Iberian Peninsula belong, discharges tend to be overestimated, which may point to the influence of reservoirs on river flow. Indeed, many Spanish stations with large errors were found to be just downstream of large dams. Meanwhile, "other regions" is a grouping of a small number of stations scattered around Europe, mainly from Finland, Italy and Iceland. Those areas, containing many

rivers in both arid and polar climates, are underrepresented in the quantification of the Bayesian Network, hence a potential reason for their lower performance.

In Fig. 5 it can be seen that the amount of scatter in the plot increases for rivers with smaller discharges. Indeed, when choosing only smaller rivers, with a catchment area of 500 km$^2$ and lower, the performance of the model drops substantially, though it still remains acceptable. Conversely, correlation or bias for stations with more than 500 km$^2$ catchment area

remains almost unchanged. We can also analyse specific river discharge, i.e. runoff divided by the respective catchment areas (Wrede et al. 2013). The $R^2$ drops to 0.52 for $Q_{AMAX}$ and 0.45 for 100-year discharge, with $I_{NSE}$ at 0.43 in both cases.

Additionally, to validate the robustness of the method, we did a split-sample test. Stations were randomly divided into two sets. Data from 917 stations were used to quantify the Bayesian Network in order to simulate discharges in the





remaining 924 stations. Of the latter, 586 stations had at least three full decades of discharge observations, which allowed us to make a comparison with simulated discharge. The validation result was almost identical with reported for the full quantification, and even a notch better: $R^2$=0.94 and $I_{NSE}$=0.93 was observed for $Q_{MAMX}$, while for 100-year discharge the same value of $I_{NSE}$ was calculated and $R^2$ equalled 0.90. Finally, the results were compared with other available studies, but

that is discussed in section 4.1.

Still, performance for individual stations varies. A selection of observed and simulated discharges, both annual maxima and fitted to Gumbel distribution, is presented in Fig. 6. In some stations, there is a very close fit, while in others either the discharge is overestimated, or the distributions have different shapes. This is however not atypical even in more local studies.

**3.2 River discharges in Europe**

Calculation of river discharges utilizing data from EURO-CORDEX climate simulations was done for years 1950–2100, and are presented here in three time slices: 1971–2000, 2021–2050 and 2071–2100. The first period is from the historical "control" run, while the other two were analysed for two emission scenarios: RCP 4.5 and RCP 8.5. Trends calculated from the data are presented in Fig. 7. For the sake of clarity, only rivers with catchment area above 500 km$^2$ are presented in the

picture; full-scale maps of discharges were included in the Supplement. Aggregate statistics by region and catchment size were included in Table 3. In the description we focus on 100-year discharge, but the trends are mostly also representative for other return periods.

The trends in Europe are very diversified. For Europe as a whole, there is a slight 4–7% increase in discharges with a 100-year return period ($Q_{100}$), with the biggest change observed in the 2021–2050 RCP 8.5 scenario. Along 34–44% of river

length in Europe, $Q_{100}$ is projected to increase at least by 10%, depending on scenario. Yet, along 16–21% a decrease by more than 10% is expected, with only small changes (±10%) for the remaining 35–49%. In RCP 8.5 both increases and decreases of $Q_{100}$ are more prominent than in RCP 4.5. In effect, $Q_{100}$ in the 2071–2100 RCP 8.5 scenario is projected to correspond to 176-year discharge under present climate (1971–2000), if we take the median value. This value is slightly lower in mid-century, and in end-century for RCP 4.5, with the smallest change compared to present climate in the 2021–

2050 RCP 4.5 scenario.

Between regions, by mid-century the largest average increases in extreme discharges are expected in the Iberian Peninsula and Danube basin (RCP 4.5), while $Q_{100}$ in Central Europe (i.e. mainly Elbe, Oder and Vistula river basins) is projected to surge even more in RCP 8.5. By the end of the century, however, Southern Europe (comprised mostly of Italy) is the region were the biggest average increase was observed in the simulations. On the other hand, $Q_{100}$ is projected to

decrease on average in the British Isles in all four scenarios, in North-East Europe (Finland, north-west Russia, the Baltics) in three scenarios, in Scandinavia in two and in South-East Europe (mainly Greece) in one. Those discrepancies are the result of several trends, namely changes in extreme precipitation, snowmelt and runoff coefficient. The first is projected to



increase across the continent, while the other two decrease at the same time, with some exceptions. Decline in snowmelt, a consequence of thinner snow cover, will contribute to lower extreme discharges in parts of Scandinavia and Scotland. However, in most of Sweden, Finland and other areas, less snowmelt twill be offset by more rainfall. Lower precipitation is expected only in small, scattered patches of Europe, most noticeably in southern Spain. At the same time, an increase of the runoff coefficient could be observed in predictions for the Iberian Peninsula and western Europe, with decreases in the remainder of the continent. Higher temperatures and less soil moisture are contributing factors to those trends.

In Table 3 trends in $Q_{100}$ were also provided per catchment size. The differences in average increase of discharges are very small, and partially caused by their uneven distribution in Europe. Median return periods are show more diversity, since relative increase in discharge by certain increment of return period typically gets smaller as the river grows in size. Most importantly, this breakdown shows that the method is able to detect trends in discharge in both large and small rivers.

## 4 Discussion

The results presented in the previous section, however encouraging on their own, have to be contrasted with other existing studies. Such analysis is presented in section 4.1, while in the subsequent subsection a discussion is carried out about the limitations of the method and the uncertainties in the model's set-up and results. Finally, ongoing and planned developments of the BN are presented.

### 4.1 Comparison with other models

The accuracy of the Bayesian Network model of extreme river discharges can be compared, directly or indirectly, with results of other physical and statistical models. In case of the former, reported values of $R^2$ and $I_{NSE}$ from several studies could be obtained. Meanwhile, the regional frequency (RFA) analysis from Smith et al. (2015) could be easily performed on our sample of European gauge stations, based on parameters provided by the authors.

Studies with measures of model performance comparable with this one where summarised in Table 4. All of them are based on LISFLOOD model, forced by a large variety of climate models. Still, the validation of this model was mainly based on Global Runoff Data Centre discharge data. Consequently, though a smaller number of gauge stations was used, they mostly overlap with the ones utilized in this study. Nevertheless, correlation between observed and simulated mean annual maxima of daily discharges ($Q_{MAMX}$), measured by $R^2$ is between 0.86 and 0.94. The corresponding value in this study is within this range. Only one study (Dankers and Feyen 2008) reported $R^2$ for discharge with different return periods ($Q_{20}$, $Q_{50}$, $Q_{100}$), and our results are slightly higher. It should be noted that in that analysis, using Gumbel distribution (like in this study) yielded better correlation than Generalized Extreme Value (GEV) distribution. Only two studies reported bias measured by $I_{NSE}$. Most interestingly, Rojas et al. (2011) show that the performance of the hydrological model changed significantly depending on how climate data were treated. The authors noted large biases in modelled precipitation data, and made a correction based on observational datasets. This modification of climate data output slightly improved the





correlation, but most importantly the $I_{NSE}$ went from a negative value, indicating poor performance, to a value close to that showing no bias at all. In this study, no modification to climate data was made and yet $I_{NSE}$ values for our statistical model are in the range of a physical model forced by bias-corrected climate data. Of course the reported validation results are not perfectly comparable due, since the described studies focused on relatively large rivers (those more than ca. 1000 km² catchment area) and used ENSEMBLES regional climate simulations, which are several years older than the CORDEX simulations employed here. Additionally, $R^2$ and $I_{NSE}$ are not the only measures available. Dankers and Feyen (2008) report that the error in simulating $Q_{MAMX}$ was bigger than 50% in 24–25% of stations and more than 100% in 6–8%. In this study, for comparable river size, i.e. with extreme discharge of ca. 100 m³/s and more, those values are 34% and 11%. Still, in overall the performance of the Bayesian Network can be described as similar to LISFLOOD model in estimating annual extremes.

To further investigate the relative accuracy of the method in light of alternate models, we performed a RFA analysis, as presented by Smith et al. (2015). This required us to obtain some supplementary data. Each river gauge station had to be assigned to one of five climate zones according to the Köppen-Geiger classification; a world map by Kottek et al. (2006) was used for that purpose. In overall, 65% of stations with long records in our sample are located in the temperate climate zone, 30% in continental, 4% in polar and 1% in arid. Additionally, mean annual rainfall was derived from CORDEX climate data. The final input information was catchment area, readily available from our datasets. In order to estimate discharge in the RFA, a given station had to be assigned to one of 82 clusters included in the RFA. The first criterion is the climate zone, which allocated a station to a group of clusters. Then, the Euclidean distance to the each cluster centroid (defined through a logarithm of area and rainfall) was calculated. Afterwards, "mean annual flood" equation (see Smith et al. 2015) was solved using the coefficients from the nearest cluster as well as catchment area and annual rainfall providing us with $Q_{MAMX}$; cluster-specific GEV distribution parameters were then applied to obtain return periods of extreme discharges.

The method provided estimates for all 1125 stations with long records, which were compared with observed discharges in Fig. 8. In case of $Q_{100}$, Gumbel-distributed discharges were used; performance with GEV distribution was lower. The performance of both BN and RFA models is visually similar, though the BN recorded higher correlation and less bias then the RFA. Less scatter can be observed in upper and lower ranges of discharges, with similar performance in the middle. Using specific river discharges the performance of both methods was lower, but still much better for the BN: $I_{NSE}$, for example, was negative for both $Q_{MAMX}$ and $Q_{100}$ when using RFA, in contrast to a value of 0.43 for the BN. RFA was devised as a global method instead of a regional one, but at the same time it is in fact a set of 82 regional approximations of hydrological processes. Here, we analyse contributing factors of extreme discharges all together, achieving comparable or better results.





## 4.2 Limitations and uncertainties

The Bayesian Network model, despite its overall high performance, has lower accuracy over some regions where outliers are observed. Some of the uncertainties and limitations of the model are immanent properties of large-scale hydrological simulations, while others are specific to how the method was conceived, and what assumptions and data were included. One of the foremost aspect belonging to the first group is that the method assumes natural flow in the catchment. Hydraulic structures, such as large dams, can have profound influence on extreme discharges, as many were developed as a flood-reducing measure. As mentioned in the results section, flows in Spanish rivers were generally overestimated, with reservoirs being a likely explanation. Continental or global scale models routinely omit this aspect, as there is not enough information available to incorporate the existence of reservoirs. They have different functions (flood protection, flow regulation, water supply) and function according to various operational procedures. The BN model includes reservoirs only indirectly; they count as lakes, and therefore contribute to the percentage of the catchment covered by water bodies, thus having negative influence of extreme discharge. However, dams can have a much larger impact on discharges, as evidenced in the lower performance of the methods in Spain, were large dams are plenty. In total, 326 large dams are within the catchments of the stations used in this study, according to the GRanD database (Lehner et al. 2008). Additionally, the conditions in the catchment may change over the timespan of the analysis of discharge data (1950–2005), due to reservoir construction or river regulation, or simply because of land use developments. Currently a single snapshot of European land cover is used (from around the year 2000), but area covered by lakes, marshes and particularly artificial surfaces is dynamic. In our analysis there was very little difference in performance between different time periods, but this aspect could be relevant locally.

The configuration of the Bayesian Network presented here was the best one we found, but may not be the only solution possible, or the best one there could be. In Paprotny and Morales Nápoles (2015) the set-up of the model was slightly different, with unconsolidated deposits (calculated a fraction of all soil types in a catchment) used instead of the runoff coefficient. It can be noticed that despite several soil datasets being mentioned in the methodology (section 2.3), none made it the final configuration of the model. Low resolution and limited thematic accuracy of global soil data is like the cause. Several other variables describing terrain, climate or land cover mentioned in section 2.3 were not included, as adding them did not improve the model. One alternative configuration, however, is worth mentioning, namely a BN incorporating terrain classification based on height-above-nearest drainage (HAND). Replacing lake and marsh cover with "wetlands" and "hillslopes" identified in the digital elevation model (see Gharani et al. 2014) caused only fractional drop in performance. Given that land cover data for Europe has very high resolution and good accuracy, this approach may give better results in areas with less satisfactory data such as the developing countries.

Some issues are related with the datasets used. Discharge data are daily values, rather than absolute peak flows, as that variable was only available from the main source of information, i.e. the Global Runoff Data Centre. Yet, Polish data were only available as sub-daily maxima, which did not affect much the accuracy for Poland or Europe, but is nonetheless a slight





inconsistency. More crucially, daily discharge is not adequate to model flash floods. These events can occur in matter of minutes, and do not even require a river bed. Also, the model utilizes daily precipitation and snowmelt, which also may not be accurate for large catchments, where the biggest floods are caused by rainfalls lasting many days. Potential incorporation different timespans of flood-inducing meteorological events is yet to be analysed. In some regions the amount of river gauge

station data was very limited, mainly in south-eastern Europe, while in others (northern and western Europe) was abundant, making the sample less representative.

Further concerns are related with the river and catchment dataset CCM2. It has lower accuracy in areas with low relief energy, otherwise known as plains. Slightest inaccuracies in the DEM result in improper delimitation of catchments in such regions. Large number lakes in post-glacial parts of Europe also result in sometimes substantial errors. For instance, $I_{NSE}$

value for $Q_{MAMX}$ for mountainous Norway is 0.90, while for Sweden, with its lake-filled landscape, it drops to 0.71. River gauge stations, for which there was a significant difference between catchment area in CCM2 and the corresponding value in the stations' metadata, were removed. The improperly divided basins still exist in our final database of simulated extreme discharges, though. This also involves omission of most artificial channels and all cases of bifurcation, river deltas included.

Climate data from CORDEX are the highest resolution available, yet biases in representing rainfall, snowmelt and

runoff could influence the results. As noticed in section 4.1, bias-correction of precipitation significantly improved performance of LISFLOOD hydrological model, therefore leaving room for further enhancements of the method. Another issue is related with climate change scenarios used to construct the database of discharges. The difference between RCP 4.5 and RCP 8.5 scenarios is sometimes very large, as witnessed in Fig. 7. This alone illustrates major uncertainty related with future projections of climate.

Finally, the underlying dependence structure requires further investigation, since some of the bivariate distributions of variables indicate that a non-Gaussian copula would a better model (see Supplemental Information for the analysis). Most prominently, the joint distribution of area and discharge shows significant tail dependence.

### 4.3 Applications and further developments

The method was originally conceived to provide extreme discharge estimates that could be used for pan-European

hazard mapping. As shown in the previous sections, it has similar accuracy to hydrological models, yet it is much faster. For hydrodynamic modelling of water levels, catchments with area greater than 100 km² where selected. In order to estimate annual maximum discharge for 246 years (56 years of the historical run and 95 years for each of the climate change scenarios) in a domain of almost 156,000 river sections above the threshold, and obtain return periods of flood event, it takes less than a day on a desktop PC. The exact value depends on the number of samples used when conditionalizing the BN, and

number of samples used to quantify the BN. Nevertheless, the method can reduce time needed to perform a flood hazard analysis, both continental-scale and local, as long as annual extremes are relevant for a particular study.



The results of this study – extreme discharges with certain return periods under present and future climate for all river sections in the domain – are publicly available online (Paprotny and Morales Nápoles 2016). It was formatted in GIS in such a manner, that it can be easily combined with the CCM2 river and catchment database. The files include a total of 10 different return periods of discharges (2–1000 years) and 5 scenarios, the same as described in section 3.2. Additionally, for

each future scenario, change in return periods of discharge with certain probability of occurrence in 1971–2000 was calculated and included in dataset. Flood hazard maps that utilized those results are also accessible; however, further discussion about them is outside the scope of this paper.

The methods scope was limited so far to Europe, but investigation is also ongoing on applying the method to other regions. Currently, data from United States and Mexico are being analysed. There is a very large number of river gauge

observations available for the contiguous US, while in its southern neighbour the number of records with good quality is limited. Mexico also lays mostly within tropical and arid climate zones, which is in stark contrast to Europe. The United States are also very geographically diversified and its biggest river system – the Mississippi-Missouri basin – is almost four times larger than the Danube basin. Moreover, for these countries global spatial datasets will be used, which have a lower resolution than those utilized in this study. This will allow us to perform a first assessment of the method's feasibility for a

global application. It is possible, for instance, to quantify the BN model with local data and analyse its performance relative to the European quantification presented in this paper, as well as combine those data.

## 5 Conclusions

In this paper we presented a first attempt to model extreme river discharges in Europe with Bayesian Networks. The method revisits the old concept of estimating discharges using only geographical properties of catchments, but with a

entirely new approach. Instead of a usual regression analysis, we determine the (conditional) correlations between different variables describing the catchments with copulas and a non-parametric BN. We show that the model has comparable accuracy to large-scale hydrological models in simulating mean annual maxima and return periods of daily discharges, and higher performance than a regional frequency analysis. The method can be applied to create basic flood scenarios at any ungauged location based on a few variables. For that reason it was used to provide estimates of extreme river discharges for

both present and future climate in all rivers in a domain covering most of the continent. Trends in discharges we found to be very diversified, while the database itself will be applied to delimiting flood hazard zones in a separate study.

The advantages of our approach is that it is has low computational expense, it is explicit and flexible. Its configuration could be easily modified, and the model can be used even if not all variables for a given location are available. At the same time it allows to perform sensitivity analysis of different variables on extreme discharges, as well as easily incorporate

changes in climate or land use over time. It purely relies on the statistical distributions and statistical dependence of catchment descriptors, without any empirical modifiers or clustering typical for other statistical methods. The aim was to make the method universal, and though it was so far only tested for Europe, the overall performance is encouraging. The





accuracy of the model changes relatively little between regions and time periods, as well as when a split-sample test is applied. The disadvantages are mostly typical for other large-scale models, such as assumption of natural flow conditions in the rivers and lower performance in smaller catchments. The method was also crafted only for annual maxima of discharges, with the purpose of accurately estimating return periods rather than discharges in a particular year. But again, this is the most relevant parameter in flood hazard analysis. The method will be further developed and tested in other parts of the world. Data availability

## Data availability

This work relied entirely on public data as inputs, which are available from the providers cited in the paper. Results of the work can be downloaded from an online repository (Paprotny and Morales Nápoles 2016).

## Acknowledgements

This work was supported by European Union's Seventh Framework Programme under "Risk analysis of infrastructure networks in response to extreme weather" (RAIN) project, grant no. 608166. The authors would like to thank the Global Runoff Data Centre in Koblenz, Germany, for kindly providing a large part of river gauge data used in this study. The work described herein benefited from useful comments by S. N. Jonkman and H. H. G. Savenije.

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





**Table 1.** Summary statistics of stations used in the work.

| Country | Number of stations | $Q_{AMAX}$ values (1950–2005) absolute | $Q_{AMAX}$ values (1950–2005) percentage | Catchment size (km$^2$) | Number of stations | $Q_{AMAX}$ values (1950–2005) absolute | $Q_{AMAX}$ values (1950–2005) percentage |
|---|---|---|---|---|---|---|---|
| France | 273 | 10642 | 14.2 | >100,000 | 32 | 1303 | 1.7 |
| Spain | 247 | 10602 | 14.2 | 10,000–100,000 | 207 | 8849 | 11.8 |
| Sweden | 283 | 10520 | 14.1 | 1000–10,000 | 513 | 20826 | 27.9 |
| United Kingdom | 228 | 9159 | 12.3 | 100–1000 | 795 | 32030 | 42.8 |
| Germany | 133 | 6996 | 9.4 | <100 | 294 | 11749 | 15.7 |
| Norway | 104 | 5035 | 6.7 | Total | 1841 | 74757 | 100.0 |
| Switzerland | 90 | 4093 | 5.5 | | | | |
| Austria | 73 | 3464 | 4.6 | | | | |
| Poland | 78 | 2807 | 3.8 | | | | |
| Finland | 53 | 2287 | 3.1 | | | | |
| Ireland | 40 | 1371 | 1.8 | | | | |
| Other countries | 239 | 7781 | 8.8 | | | | |

**Table 2.** Validation results for simulated and observed average annual maxima of daily river discharges $Q_{MAMX}$ and annual maxima with a 100-year return period $Q_{100}$.

| Category | | Stations | $Q_{MAMX}$ $R^2$ | $Q_{MAMX}$ $I_{NSE}$ | $Q_{MAMX}$ $I_{RSR}$ | $Q_{100}$ $R^2$ | $Q_{100}$ $I_{NSE}$ | $Q_{100}$ $I_{RSR}$ |
|---|---|---|---|---|---|---|---|---|
| | Total | 1125 | 0.92 | 0.92 | 0.29 | 0.89 | 0.80 | 0.44 |
| Regions | Central Europe | 138 | 0.89 | 0.71 | 0.54 | 0.86 | 0.85 | 0.39 |
| | British Isles | 145 | 0.86 | 0.85 | 0.39 | 0.81 | 0.77 | 0.48 |
| | Western Europe | 261 | 0.97 | 0.96 | 0.19 | 0.94 | 0.79 | 0.46 |
| | Iberian Peninsula | 112 | 0.79 | 0.78 | 0.47 | 0.71 | 0.57 | 0.65 |
| | Danube basin | 167 | 0.93 | 0.92 | 0.27 | 0.92 | 0.83 | 0.42 |
| | Scandinavia | 227 | 0.92 | 0.83 | 0.42 | 0.91 | 0.90 | 0.31 |
| | Other regions | 75 | 0.79 | 0.82 | 0.43 | 0.72 | 0.70 | 0.55 |
| Time period | 1951–1980 | 512 | 0.93 | 0.92 | 0.28 | 0.89 | 0.85 | 0.38 |
| | 1961–1990 | 792 | 0.93 | 0.92 | 0.28 | 0.90 | 0.85 | 0.39 |
| | 1971–2000 | 958 | 0.93 | 0.93 | 0.27 | 0.90 | 0.84 | 0.40 |
| | 1981–2010 | 765 | 0.91 | 0.91 | 0.31 | 0.87 | 0.83 | 0.42 |
| Catchment area | >500 km$^2$ | 605 | 0.92 | 0.91 | 0.30 | 0.88 | 0.78 | 0.47 |
| | <500 km$^2$ | 520 | 0.59 | 0.52 | 0.69 | 0.56 | 0.55 | 0.67 |
| | Specific discharge | 1125 | 0.52 | 0,43 | | 0.45 | 0.43 | |

**Table 3.** Projected change in 100-year river discharge ($Q_{100}$) relative to 1971–2000, and return periods of discharge equal to $Q_{100}$ in 1971–2000.

| Category | Average change in $Q_{100}$ weighted by | Median return period of discharge |
|---|---|---|



| | | length of river sections (%) | | | | equal to $Q_{100}$ in 1971–2000 (years) | | | |
|---|---|---|---|---|---|---|---|---|---|
| | | 2021–2050 RCP4.5 | 2071–2100 RCP4.5 | 2021–2050 RCP8.5 | 2071–2100 RCP8.5 | 2021–2050 RCP4.5 | 2071–2100 RCP4.5 | 2021–2050 RCP8.5 | 2071–2100 RCP8.5 |
| | Total | +3.7 | +5.7 | +7.0 | +5.9 | 133 | 168 | 163 | 176 |
| Regions (selected) | Central Europe | +3.5 | +9.6 | +13.5 | +12.2 | 138 | 200 | 225 | 276 |
| | British Isles | -6.0 | -6.5 | -6.8 | -13.5 | 59 | 62 | 58 | 42 |
| | Southern Europe | +3.9 | +12.1 | +8.8 | +17.7 | 142 | 311 | 209 | 492 |
| | Western Europe | +1.1 | +4.5 | +5.8 | +11.4 | 116 | 163 | 174 | 269 |
| | Iberian Peninsula | +7.3 | +8.1 | +12.2 | +11.0 | 181 | 177 | 215 | 206 |
| | Danube basin | +6.5 | +9.4 | +9.3 | +8.0 | 173 | 234 | 190 | 207 |
| | North-East Europe | +1.2 | -0.1 | -1.5 | -8.4 | 99 | 117 | 87 | 64 |
| | Scandinavia | +1.8 | -1.9 | +4.6 | -5.0 | 121 | 110 | 184 | 80 |
| | South-East Europe | +1.2 | +2.7 | -1.2 | +3.7 | 137 | 135 | 111 | 149 |
| Catchment area | >100,000 km$^2$ | +2.9 | +6.4 | +8.2 | +5.2 | 195 | 500 | 685 | 337 |
| | 10,000–100,000 km$^2$ | +4.7 | +7.4 | +8.9 | +7.2 | 168 | 205 | 269 | 227 |
| | 1000–10,000 km$^2$ | +3.3 | +4.3 | +6.0 | +5.1 | 133 | 156 | 173 | 162 |
| | 100–1000 km$^2$ | +3.7 | +5.1 | +5.7 | +5.7 | 128 | 163 | 170 | 159 |
| | <100 km$^2$ | +2.9 | +4.4 | +3.8 | +5.0 | 134 | 170 | 162 | 178 |

**Table 4.** Reported validation results for extreme discharge simulations for Europe.

| Study | Description | | Variable | Measure | |
|---|---|---|---|---|---|
| | | | | $R^2$ | NSE |
| This study | Bayesian Network model, 1125 stations | | $Q_{MAMX}$ | 0.92 | 0.92 |
| | | | $Q_{100}$ | 0.89 | 0.80 |
| Dankers and Feyen (2008) | LISFLOOD model, 2 different climate model resolutions, 1961–1990, 209 stations, Gumbel or GEV distribution | | $Q_{MAMX}$ | 0.90–0.91 | - |
| | | | $Q_{100}$ | 0.80–0.87 | - |
| | | | $Q_{50}$ | 0.84–0.88 | - |
| | | | $Q_{20}$ | 0.86–0.88 | - |
| Dankers and Feyen (2009) | LISFLOOD model, 8 different climate models and runs, 1961–1990, 209 stations | | $Q_{MAMX}$ | 0.86–0.93 | - |
| Rojas et al. (2011) | LISFLOOD model, 1961–1990, 554 stations | Without bias correction of climate data | $Q_{MAMX}$ | 0.87 | -1.89 |
| | | With bias correction | $Q_{MAMX}$ | 0.92 | 0.89 |
| Rojas et al. (2012) | LISFLOOD model, 12 different bias-corrected climate models, 1961–1990, 554 stations | | $Q_{MAMX}$ | 0.90–0.94 | 0.88–0.93 |





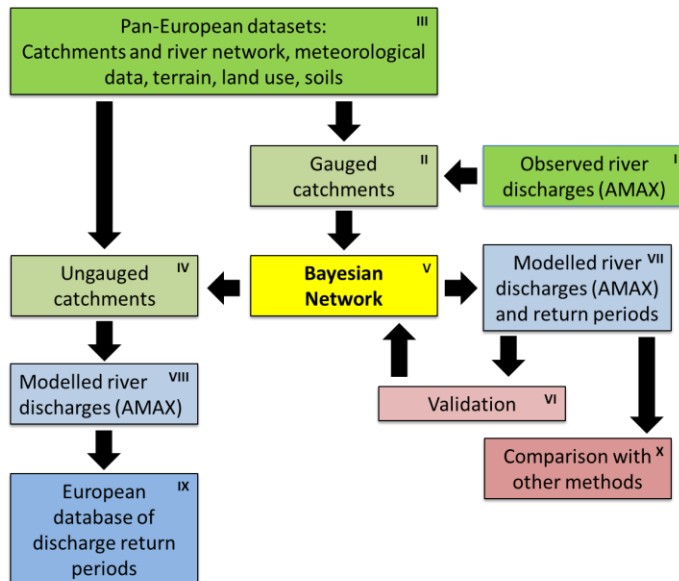

**Figure 1.** Schematic workflow of obtaining extreme river discharges from catchment characteristics. $Q_{AMAX}$ = annual maxima of discharges. Roman numerals refer to the text.





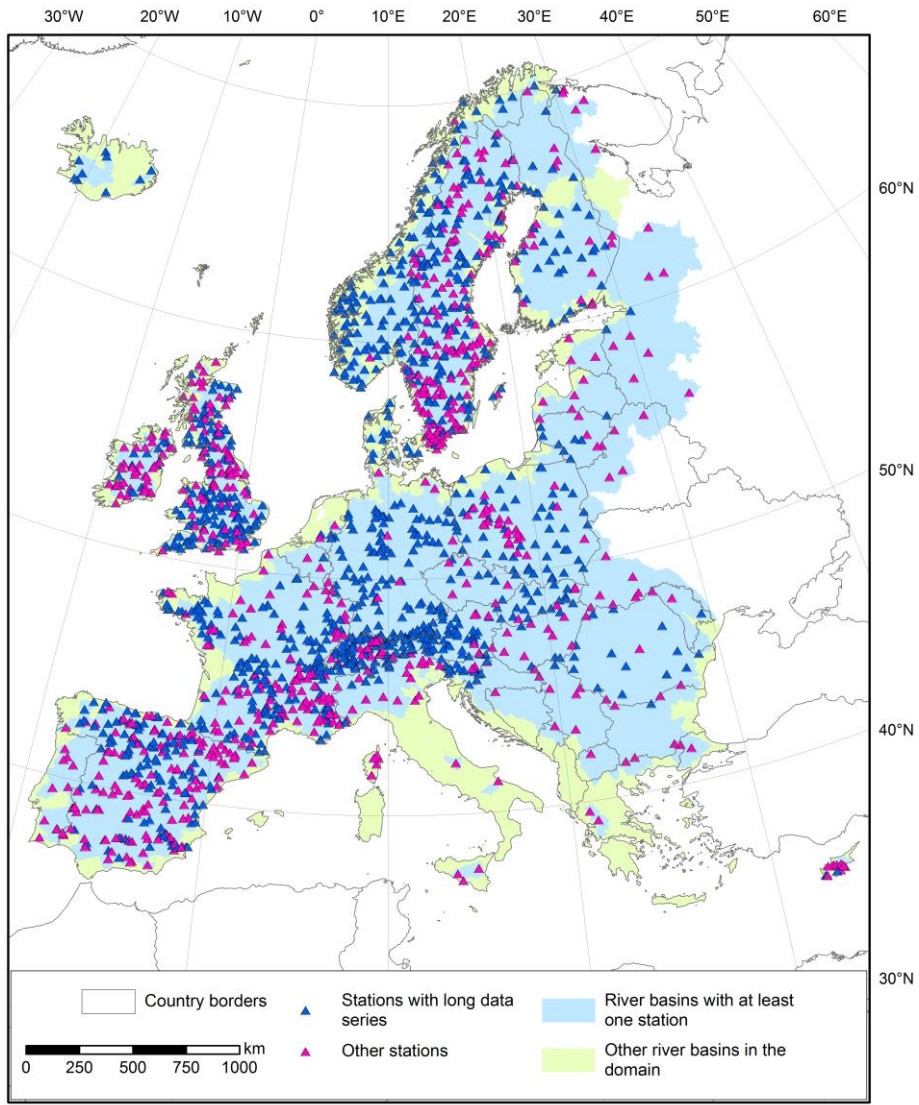

**Figure 2.** Measurement stations used in the work ("long data series" indicates stations with sufficient data for calculating return periods) and river basins included in the domain.



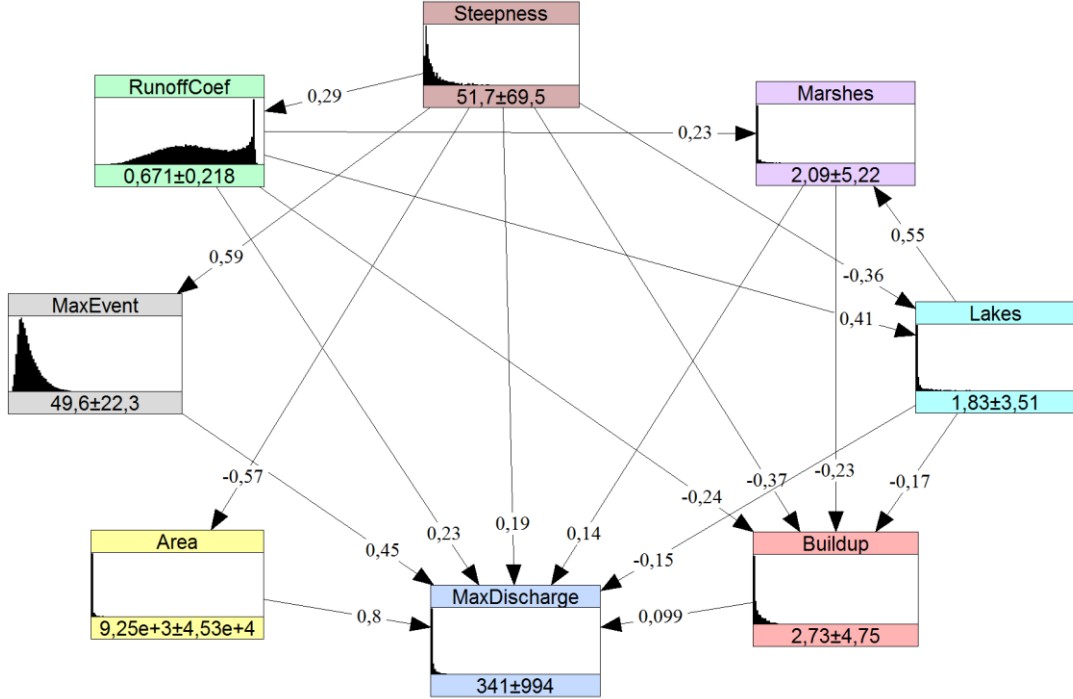

**Figure 3.** Bayesian Network for river discharges in Europe. The nodes are presented as histograms, with numbers indicating the means and standard deviations of the variables. Values on the arcs are the (conditional) rank correlation coefficients.

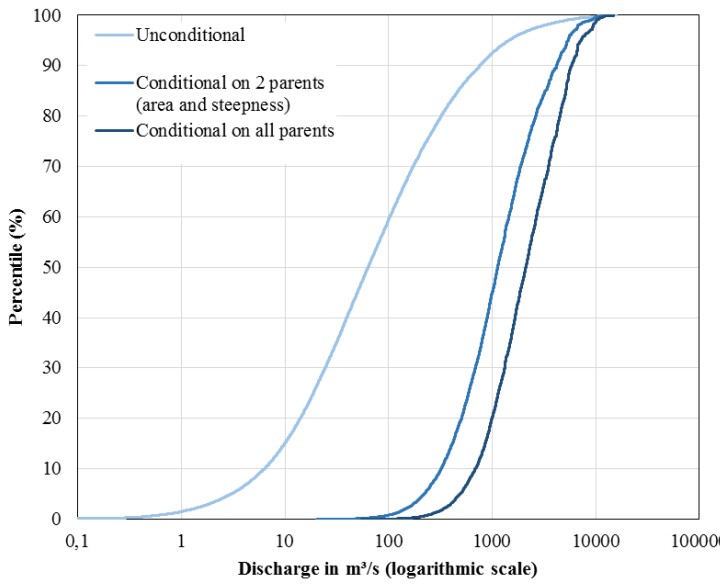

**Figure 4.** Cumulative probability distribution of river discharge: unconditional and conditionalized on two and seven nodes using values for Basel station in Switzerland (river Rhine, year 2005).





(a)                                                             (b)



5                                      (c)                                                             (d)

**Figure 5.** Simulated and observed average annual maxima of daily river discharges (a) and annual maxima fitted to Gumbel distribution to calculate 1000-, 100- and 10-year return periods (b–d), for 1125 station. 30-year periods of annual maxima were used (the most recent

10    available out of 1971–2000, 1961–1990 or 1951–1980).





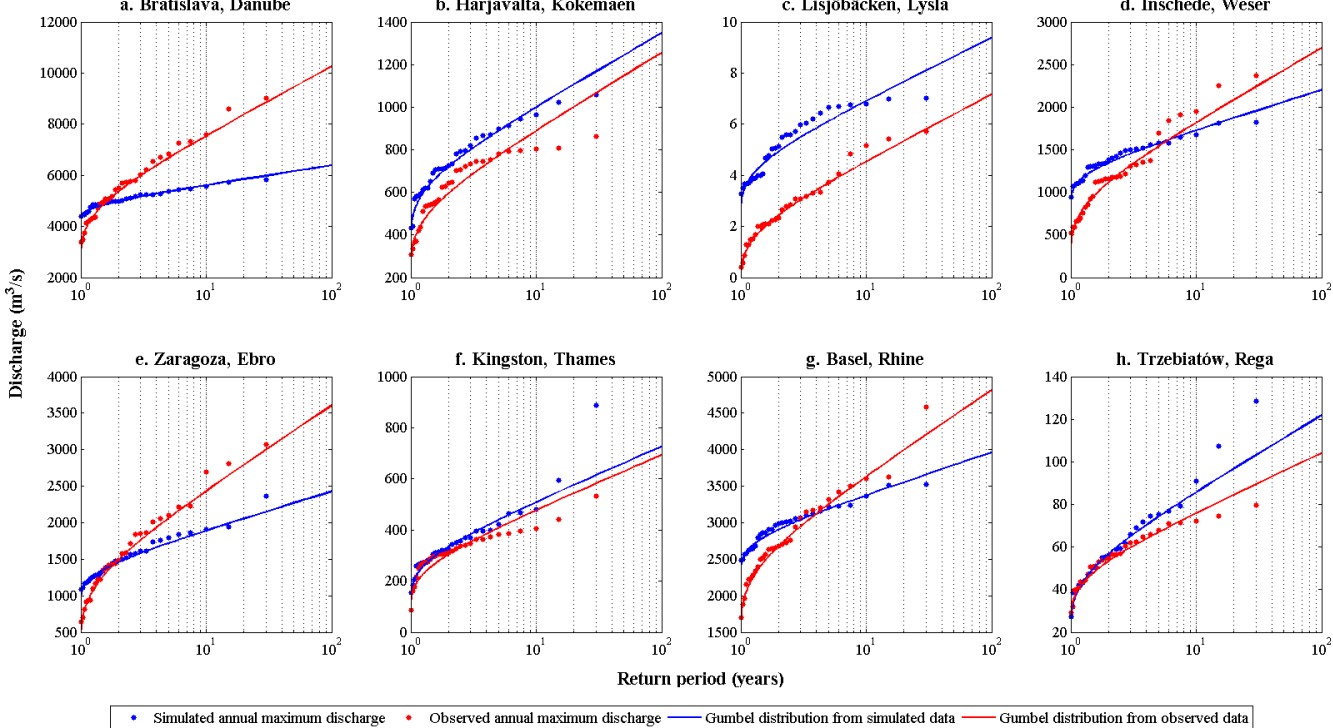

**Figure 6.** Simulated and observed annual maxima of daily river discharges fitted to Gumbel distribution at selected stations. Data refer to 1971–2000, except h), which refers to 1961–1990.





**Figure 7.** Predicted trends in daily river discharge with a 100-year return period (Gumbel distribution) under climate change scenarios (rivers with catchment area above 500 km$^2$ only).





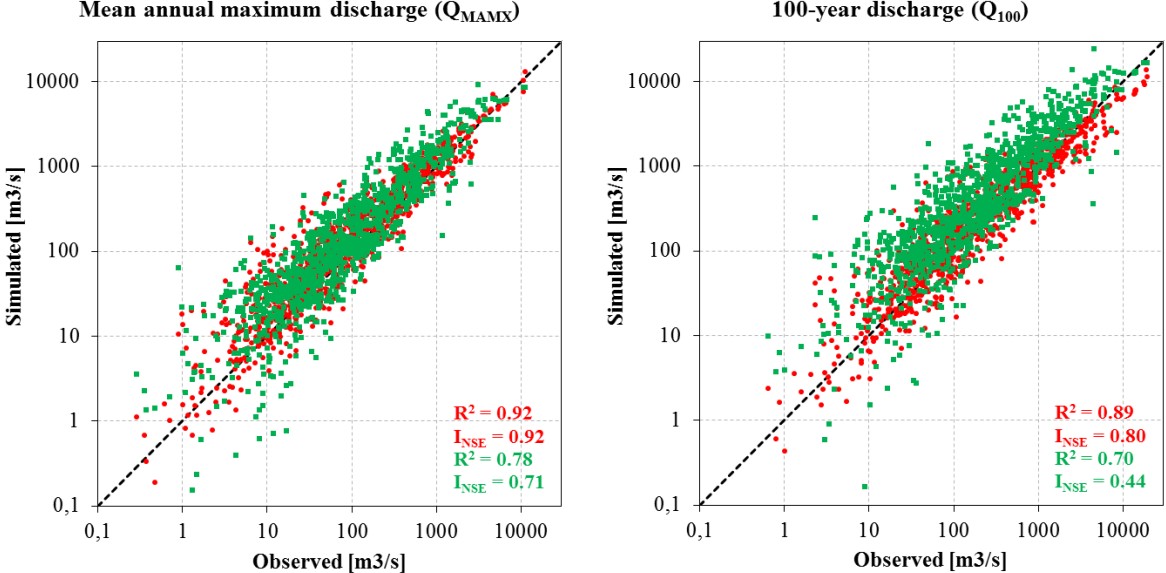

**Figure 8.** Simulated and observed average annual maxima of daily river discharges and 100-year discharge for 476 stations; Bayesian Network model in red, regional frequency analysis in green. 30-year periods of annual maxima were used (the most recent available out of 1971–2000, 1961–1990 or 1951–1980).