# Peer review of "Estimating extreme river discharges in Europe through a Bayesian Network"

_Hydrology and Earth System Sciences, 2016_

## Referee Comment (RC1) · Anonymous Referee #1 · 26 Jul 2016

The paper combines Copulas with the Bayesian network framework aiming to construct a precise multivariate model for extreme river discharges. The approach is very challenging and requires a good understanding of both methodological concepts, Bayesian networks and Copulas. The paper is well structured and organized, especially considering the bunch of information and background knowledge that needs to be explained/referred to. Yet, some proceedings/issues are not clear to me.

section 2.4:

The choice or the Gaussian copula could be better justified, e.g. are there any physical explanations?

[Figure]

- Mention in the main text the two types of copulas you are testing as well.

- Briefly explain in the supplement, why you choose the selected types for comparison and not other/further representatives of no/lower/upper tail dependence. Are there any physical explanations? The test only shows, that Gaussian performs better than the gumbel and clayton, but it might still be a bad choice. Supplement 3 even indicates that the Gaussian copula is not a very good choice.

I do not understand how you get and use the conditional rank correlations for continuous distributions. The references you provide only give a short explanation and refer to further literature again. Is there some standard literature on the definition of conditional rank correlations for continuous distributions? To my understanding the conditional rank correlation depends on the state of additional parents, yet in fig. 3 you only give single numbers (no conditioning is visible). Yet, if the rank correlation is independent of the states of the other parents, you miss to model the joint effects and could use a naive Bayes instead (which would be far more simple than going all the way over BNs and copulas). On the other hand, if the conditional rank correlation depends on other parents states, there must be a way to calculate it for each possible conditioning state (to be able to perform inference) or it must be determined for a discrete approximation of the conditioning variable (which increases the number of required parameters significantly; similar as for using discrete BNs from the beginning). Maybe you could comment on this in the discussion forum.

section 2.6:

I find this subsection difficult to read and suggest to revise the sentence structure of this section.

Why do you use 30-year time periods? What would be the effect of using shorter/longer time periods?

Please mention the distributions you are testing in addition to GEV. GEV performs best

compared to what?

section 3.1:

p.13, l. 5: "2-year discharge has the same performance as Q_MAMX" <- where does this information come from?

Why do you suddenly have 4 different time periods? Section 2.6 mentions only 3.

The regional performance seems to depend strongly on the number of stations used per region

You mention several times 'the model performance remains acceptable'. What is your understanding of acceptable?

How do you explain the better performance of the BN quantified from the smaller dataset of 917 records?

p. 18:

You might extend your comment on using different types of copulas: How suitable is the Gaussian to model all interactions? How well does it fit the data (are there objective measures)? Would it be possible to use different types of copulas in the same BN and thus find a better description of each interaction? Which other types of copulas could be useful to check? What do you expect, to which extend could the model be improved, by using different types of copulas?

figure 3 and supplement 4:

Why do you use a discrete BN in the shown examples for inference? This does not correspond with your objective to find/use a continous BN with a low number of parameters. The discrete conditional distributions you show in the supplement, are not smooth. I guess, this should not happen, if you stick to continuous representations.

Typos and minor issues:

p. 6, l.30: the influence of DIFFERENCES models

p. 8, l.14: need to explained

p. 8, l.22: a set of noTes and arcs

p. 9, l. 1: which is the actually case

p. 9, l. 3: to be precise, the actual number of required conditional probabilities/parameters is a bit smaller, since some parameters result from probability theory (the parameters that describe a distribution for a specific condition have to sum up to one)

p. 9, l.28: Hanea 2006 is missing in the list of references

p.10, l.27: values these climate variables

p.11, l. 9: This variable is influence

p.11, l.11-12: check complete sentence

p.11, l.21: allowed to performed

p.15, l. 8: median return periods are show

p.17, l.12: I would not consider this fact as "evidenced", but rather as indicated

p.18, l. 3: Potential incorporation different time spans

p.18, l.21: non-Gaussian copula would a better model

p.20, l. 6

---

## Referee Comment (RC2) · Anonymous Referee #2 · 9 Aug 2016

The manuscript used a new Bayesian network to estimate extreme river discharges, which incorporate climate and large-scale spatial environmental datasets, and the joint distribution of the environmental variables are quantitatively considered. In general, the manuscript is well written, the methodology is well presented and the results are consistent. I am supportive of the paper, while having the following questions and comments.

1. page 9, line 5. It is not clear to me which empirical marginal distribution is used for the variables considered in Fig3. In particular, I am curious on which marginal distribution is used for Max discharge data. Whether it is an extreme value distribution (or skewed distribution) or some other distribution? Please explain the reason if just a

Normal distribution is used, because the extreme data may not fit well.

2. Page 5, line 3. (VII). From the Bayesian network, a conditional/marginal distribution of the max discharge data can be obtained. But how a particular value of the annual maxima is obtained? Is it the median?

3. Section 2.5 and Fig 3. Seven variables are considered as parents of max discharge in the manuscript, which means that the max discharge is "predicted" using 7 variables. Since the network is not created automatically, manually developing a model with seven predictors could cause the problem of overfitting (a good fit but with pool predictability). The model validation (section 3.1) showed the fit is good, but the "risk" of overfitting is not discussed. Usually a cross-validation need to be considered, but I understand that it is hard to conduct such analysis for the max discharge with limited data. I think it may worth to put some discussion of overfitting in the manuscript.

4. Some references are missing, such as Hanea et al. 2006 (page 9, line 28) and Mutua 1994 (page 11 line 26). Please check carefully for the rests.

---

## Author Comment (AC2) · 25 Aug 2016

We would like thank the referee for the time spent in reviewing our article and the valuable comments. Below, we list all the comments and our response.

Page 5, line 3. (VII). From the Bayesian network, a conditional/marginal distribution of the max discharge data can be obtained. But how a particular value of the annual maxima is obtained? Is it the median?

-> As we mention in section 2.5, page 11, we calculate the mean of the conditional probability distribution of annual maxima of discharge.

Page 9, line 5. It is not clear to me which empirical marginal distribution is used for the

variables considered in Fig3. In particular, I am curious on which marginal distribution is used for Max discharge data. Whether it is an extreme value distribution (or skewed distribution) or some other distribution? Please explain the reason if just a Normal distribution is used, because the extreme data may not fit well.

-> The distributions are all empirical, i.e. non-parametric. We use the usual estimator of the cumulative probability distribution; please see the supplement to this reply for the equation. The conditional probability distribution of annual maxima of discharge is empirical as well. We will clarify this in the text. No marginal Normal distribution is used; it is only the joint distribution that is modelled through a Normal, or Gaussian, copula.

Section 2.5 and Fig 3. Seven variables are considered as parents of max discharge in the manuscript, which means that the max discharge is "predicted" using 7 variables. Since the network is not created automatically, manually developing a model with seven predictors could cause the problem of overfitting (a good fit but with pool predictability). The model validation (section 3.1) showed the fit is good, but the "risk" of overfitting is not discussed. Usually a cross-validation need to be considered, but I understand that it is hard to conduct such analysis for the max discharge with limited data. I think it may worth to put some discussion of overfitting in the manuscript.

-> By standards of Bayesian Networks, a model with 8 nodes is relatively small. One way to verify overfitting is by testing the model in other geographical zones. It should be noted that in the manuscript we perform a split-sample validation and achieved the same results. We will amend the last paragraph of the discussion to mention overfitting.

Some references are missing, such as Hanea et al. 2006 (page 9, line 28) and Mutua 1994 (page 11 line 26). Please check carefully for the rests.

-> We thank the reviewer for pointing us to this error. List of references will be revised and corrected; we identified two more errors. The missing citations are:

- Hanea, A. M., Kurowicka, D., Cooke, R. M.: Hybrid Method for Quantifying and Analyzing Bayesian Belief Nets, Qual. Reliab. Engng. Int., 22, 709–729, doi:10.1002/qre.808, 2006.

- Moriasi, D., Arnold, J., Van Liew, M., Binger, R., Harmel, R., and Veith T.. Model evaluation guidelines for systematic quantification of accuracy in watershed simulations. T. ASABE, 50, 885–900, 2007.

- Mutua, F. M.: The use of the Akaike Information Criterion in the identification of an optimum flood frequency model, Hydrolog. Sci. J., 39, 235–244, doi:10.1080/02626669409492740, 1994.

Please also note the supplement to this comment:
http://www.hydrol-earth-syst-sci-discuss.net/hess-2016-250/hess-2016-250-AC2-supplement.pdf
* * *
[Figure]

**Supplement:**

Estimator of the cumulative probability distribution:

$$\hat{F}(x) = \frac{1}{n} \sum_{i=1}^{n} 1_{\{x_i \leq x\}}$$ (1)

where $(x_i, ..., x_n)$ are the samples of a random variable.

---

## Editor Comment (EC1) · J. Seibert (Editor) · 13 Sep 2016

Thanks for submitting this interesting manuscript to HESS. This manuscript presents a new approach to discharge peak estimation, which obviously is a very timely issue. The reviewers provide valuable comments on the manuscript. In addition I have the following concerns:

The presentation of the BN performance in figs 5&8 is misleading as the values are given in m3/s, where results always look much nicer than if specific discharge values would be plotted (see figs 6&7 in Wrede et al. (2013) as example). Based on my comments &reply before publication in HESS-D, I understand the aim to compare results with previous papers (which used the m3/s comparison) and the drop in performance

is now mentioned in one sentence. However, it would be much better if the plots would present the simulations in the fair way (i.e. specific discharge values) and the comparison would be given in words that vice versa. After all, we should aim at providing examples of good practices!

Opposite to the authors' interpretation of Rojas et al. (2012) different climate models (GCM/RCM combinations) can provide largely varying results. Therefore, the use of only one GCM/RCM comination is a clear limitation and largely ignoring the uncertainties caused by the climate models (see conclusions in Rojas et al. 2012; and recent papers by PhD students in my group (Teutschbein and Seibert, 2012; Addor et al., 2014). Furthermore, given the often significant biases in climate model simulations, the decision to not use any bias correction is surprising at least. Both decisions (only one model, no bias correction) need to be motivated more convincingly. At least I would recommend the authors to test the effects of using another model and/or bias correction.

My major concern, however, is figure 7, which illustrates predicted future changes in the 100-year floods over Europe. Such figures easily get the attention of media and decision makers and as scientists we, thus, really have to be careful in how we communicate such results. Presenting such results without also quantifying uncertainties is problematic. I, thus, strongly recommend to quantify the uncertainties of the results shown in fig. 7. There are partly huge differences in the predicted changes in neighboring streams (even in regions where snow is not important) and I am missing an explanation of these differences. One approach to help understanding the patterns would be to use a uniform precipitation change, so that the effects of differences in catchment characteristics versus differences in precip-changes could be disentangled. (In addition, as far as I can see, the climate change prediction is not mentioned in the methods part.)

Addor, N., Rössler, O., Köplin, N., Weingartner, R., Seibert, J., 2014. Robust changes and sources of uncertainty in the projected hydrological regimes of Swiss catchments,

[Figure]

Water Resources Research, 50(10), 7541–7562, doi:10.1002/2014WR015549.

Teutschbein, C., and Seibert, J., 2012. Bias Correction of Regional Climate Model Simulations for Hydrological Climate-Change Impact Studies: Review and Evaluation of Different Methods. Journal of Hydrology, 456-457:11-29. doi:10.1016/j.jhydrol.2012.05.052.

Wrede, S., J. Seibert, and S. Uhlenbrook. 2013. Distributed conceptual modelling in a Swedish lowland catchment: a multi-criteria model assessment. Hydrology Research 44(2):318-333. Doi: Doi 10.2166/Nh.2012.056.

---

## Author Response (AR1)

We would like thank the referees and the editor for the time spent in reviewing our article and their valuable comments. All the comments and observations have contributed to a significant improvement of the presentation of our study. Below, we list all the comments and our response, and provide the marked-up version of the manuscript afterwards.

*Section 2.4: The choice or the Gaussian copula could be better justified, e.g. are there any physical explanations?*
There are no physical explanations; the choice of the type of copula is based purely on the dependency structure.

*Mention in the main text the two types of copulas you are testing as well.*
Names of the other copulas (Gumbel, Clayton) were added to the main text.

*Briefly explain in the supplement, why you choose the selected types for comparison and not other/further representatives of no/lower/upper tail dependence. Are there any physical explanations? The test only shows, that Gaussian performs better than the gumbel and clayton, but it might still be a bad choice. Supplement 3 even indicates that the Gaussian copula is not a very good choice.*
The three copulas used in the study have, primarily, the advantage of having one parameter which is obtained directly from the observational data. In this way, the method is more flexible. Naturally, the choice is not perfect, though this is not different to numerous possibilities in regression or extreme value analyses. These copulas cover basic asymmetries from data: upper and lower dependence. Since this is a first approximation of the joint distribution pointing to possible asymmetries observed in the data was considered sufficient. Future research could be directed to exploring other asymmetries in bivariate distributions in which case the copula families to be tested should be larger. Comments on this direction have been added to the paper. For example, in section 4.2 we have added the following: "Other copulas could potentially be used, as for some distributions tail dependence and other asymmetries may be present. Even though the normal copula works well most of the time. Skewness for example may be modelled by copulas based on mixture distributions. This would correspond to copulas with more than two parameters (Joe 2014)."

*I do not understand how you get and use the conditional rank correlations for continuous distributions. The references you provide only give a short explanation and refer to further literature again. Is there some standard literature on the definition of conditional rank correlations for continuous distributions? To my understanding the conditional rank correlation depends on the state of additional parents, yet in fig. 3 you only give single numbers (no conditioning is visible). Yet, if the rank correlation is independent of the states of the other parents, you miss to model the joint effects and could use a naive Bayes instead (which would be far more simple than going all the way over BNs and copulas). On the other hand, if the conditional rank correlation depends on other parents states, there must be a way to calculate it for each possible conditioning state (to be able to perform inference) or it must be determined for a discrete approximation of the conditioning variable (which increases the number of required parameters significantly; similar as for using discrete BNs from the beginning). Maybe you could comment on this in the discussion forum.*
Conditional rank correlations are calculated as shown in eq. 4, except that the conditional distributions are used inside the arguments to the right of the equal sign. For the Gaussian copula conditional correlations are equal to partial correlations and these are constant. Hence the importance of validating the choice of the Gaussian copula. We thank the referee for noticing this fact, we made it more evident in the text. The full explanation could be found in the publication we refer to in that paragraph – Hanea et al. 2015; it contains the theorems (3.1, 3.2) as well as the proof (Appendix A). That proof shows that in a non-parametric Bayesian Network the joint distribution is uniquely determined. Therefore, the correlations shown in Fig. 3 are all conditional on their parents. As we note further in the methodology in the main text, conditioning is done by sampling the BN. This procedure is shown by Hanea et al. 2006; unfortunately, the reference got lost in the original submission:
Hanea, A. M., Kurowicka, D., Cooke, R. M.: Hybrid Method for Quantifying and Analyzing Bayesian Belief Nets, Qual. Reliab. Engng. Int., 22, 709–729, doi:10.1002/qre.808, 2006.
We add in section 2.4: "Conditional rank correlations are calculated as shown in eq. 4, except that the conditional distributions are used inside the arguments to the right of the equal sign. For the Gaussian copula conditional correlations are equal to partial correlations and these are constant."

*Section 2.6: I find this subsection difficult to read and suggest to revise the sentence structure of this section.*

The section was revised and reorganized for clarity.

*Why do you use 30-year time periods? What would be the effect of using shorter/longer time periods?*
30-year periods were used, because 1) it maximises the number of stations for validation and 2) it is the most commonly used time period in climate research, and was also used in the research project for which this method was developed. This is now indicated in section 2.6. We should also note, that if we use a 50-year period (1951–2000), we can only validate the model on 378 stations (one-third of the current validation data set). Yet, the results for $Q_{100}$ calculated from 50 years of data are: $R^2 = 0.92$ and NSE = 0.90, which is slightly better than using 30 years of data.

*Please mention the distributions you are testing in addition to GEV. GEV performs best compared to what?*
There were 15 other distributions analysed; we now mention a few most relevant in the text (generalized Pareto, gamma, lognormal or Weibull)

*Section 3.1, p.13, l. 5: "2-year discharge has the same performance as Q_MAMX" <- where does this information come from?*
To align better the text to the graphs, we corrected this information to "10-year discharge has almost the same performance as Q_MAMX"

*Why do you suddenly have 4 different time periods? Section 2.6 mentions only 3.*
The 1981–2010 was indeed not mentioned earlier. It was only used to analyse the results, and not to produce the final database. A mention of it was added in section 2.6.

*The regional performance seems to depend strongly on the number of stations used per region.*
We do not believe there is a link between regional performance and number of stations. For instance, we can break down further the regions into countries, and this shows that, when comparing modelled and observed Q_MAMX, e. g. Sweden (139 stations) has $R^2$ of 0.71, while Norway (101 station) has 0.90, which in case of the former is worse than the value for Scandinavia, and the latter is better. Finland (36 stations) has $R^2$=0.88, and included in "other regions" (75 stations), where $R^2$ equals 0.79. We added this information to section 4.2.

*You mention several times 'the model performance remains acceptable'. What is your understanding of acceptable?*
We indeed use 'acceptable' two times in the text. We used Moriasi et al. (2007) as a reference. Information about this was added to the text.

*How do you explain the better performance of the BN quantified from the smaller dataset of 917 records?*
The difference is at most 0.01 in $R^2$ or NSE, so it is really negligible.

*Page 18: You might extend your comment on using different types of copulas: How suitable is the Gaussian to model all interactions? How well does it fit the data (are there objective measures)? Would it be possible to use different types of copulas in the same BN and thus find a better description of each interaction? Which other types of copulas could be useful to check? What do you expect, to which extend could the model be improved, by using different types of copulas?*
We now mention more explicitly the aspects touched in the comment, which are addressed in other places in the manuscript and especially in the supplement. Briefly:
1) We test the joint normal copula assumption using the determinant of the rank correlation matrix (sec. 3 of the supplement);
2) We show the statistical analysis of the results in section 3.1 and 4.1, concluding that the fit is good in contrast to other studies and methods;
3-5) In section 4.2 we now write: "Other copulas could potentially be used, as for some distributions tail dependence and other asymmetries may be present. Even though the normal copula works well most of the time. Skewness for example may be modelled by copulas based on mixture distributions. This would correspond to copulas with more than two parameters

[Joe 2014]." There is surely potential for improvements by more detailed analysis of the dependency structures in future research we now highlight this fact in the discussion section.

*Figure 3 and supplement 4:* *Why do you use a discrete BN in the shown examples for inference? This does not correspond with your objective to find/use a continuous BN with a low number of parameters. The discrete conditional distributions you show in the supplement, are not smooth. I guess, this should not happen, if you stick to continuous representations.*
Empirical distributions are not smooth. A discrete BN uses conditional probability tables, while the class of continuous BN we use require empirical one-dimensional margins and rank correlations. The visual impression that the marginal distributions are discrete is merely caused by the representation of the marginal empirical distributions as histograms, as noted below Figure 3.

*Page 9, l. 3:* *to be precise, the actual number of required conditional probabilities/parameters is a bit smaller, since some parameters result from probability theory (the parameters that describe a distribution for a specific condition have to sum up to one)*
We thank the reviewer for pointing it out; the following remark was added to the text:

"If a node has 7 parents, as it happens in the BN described in the next section, and it is discretized into 5 states, a probability table with $5^8 = 390,625$ conditional probabilities would be required. $5^7 = 78,125$ may be estimated by difference, as probabilities must add to 1. Thus, 312,500 probabilities would need to be specified. Similarly, if we were to discretize into 10 states each node 90,000,000 probabilities would need to be specified. Even a discretization into 5 states for each node in our model would make the quantification prohibitive given the data available. Considering other nodes, which also have parents, would make it even more restrictive for the use of discrete BNs."

*Typos and minor issues*
*p. 6, l.30: the influence of DIFFERENCES models*
*p. 8, l.14: need to explained*
*p. 8, l.22: a set of noTes and arcs*
*p. 9, l. 1: which is the actually case*
*p. 9, l.28: Hanea 2006 is missing in the list of references*
*p.10, l.27: values these climate variables*
*p.11, l. 9: This variable is influence*
*p.11, l.11-12: check complete sentence*
*p.11, l.21: allowed to performed*
*p.15, l. 8: median return periods are show*
*p.17, l.12: I would not consider this fact as "evidenced", but rather as indicated*
*p.18, l. 3: Potential incorporation different time spans*
*p.18, l.21: non-Gaussian copula would a better model*
*p.20, l. 6*
We thank the reviewer for the detailed listing of smaller mistakes. All listed typos were fixed in the manuscript.

**Referee #2**

*Page 5, line 3. (VII). From the Bayesian network, a conditional/marginal distribution of the max discharge data can be obtained. But how a particular value of the annual maxima is obtained? Is it the median?*
As we mention in section 2.5, page 11, we calculate the mean of the conditional probability distribution of annual maxima of discharge.

*Page 9, line 5. It is not clear to me which empirical marginal distribution is used for the variables considered in Fig3. In particular, I am curious on which marginal distribution is used for Max discharge data. Whether it is an extreme value*

*distribution (or skewed distribution) or some other distribution? Please explain the reason if just a Normal distribution is used, because the extreme data may not fit well.*

The distributions are all empirical, i.e. non-parametric. We use the usual estimator of the cumulative probability distribution:

$$\hat{F}(x) = \frac{1}{n} \sum_{i=1}^{n} 1_{\{x_i \leq x\}}$$

where $(x_i, ..., x_n)$ are the samples of a random variable. The conditional probability distribution of annual maxima of discharge is empirical as well. We now clarify this in the text. No marginal Normal distribution is used; it is only the joint distribution that is modelled through a Normal, or Gaussian, copula. We thank the reviewer for this observation.

***Section 2.5 and Fig 3.*** *Seven variables are considered as parents of max discharge in the manuscript, which means that the max discharge is "predicted" using 7 variables. Since the network is not created automatically, manually developing a model with seven predictors could cause the problem of overfitting (a good fit but with pool predictability). The model validation (section 3.1) showed the fit is good, but the "risk" of overfitting is not discussed. Usually a cross-validation need to be considered, but I understand that it is hard to conduct such analysis for the max discharge with limited data. I think it may worth to put some discussion of overfitting in the manuscript.*

By standards of Bayesian Networks, a model with 8 nodes is relatively small. One way to verify overfitting is by testing the model in other geographical zones. It should be noted that in the manuscript we perform a split-sample validation and achieved the same results. We amended the last paragraph of the discussion to mention overfitting.

*Some references are missing, such as Hanea et al. 2006 (page 9, line 28) and Mutua 1994 (page 11 line 26). Please check carefully for the rests.*

We thank the reviewer for pointing us to this error. List of references was revised and corrected; we identified two more errors. The missing citations are:

Hanea, A. M., Kurowicka, D., Cooke, R. M.: Hybrid Method for Quantifying and Analyzing Bayesian Belief Nets, Qual. Reliab. Engng. Int., 22, 709–729, doi:10.1002/qre.808, 2006.

Moriasi, D., Arnold, J., Van Liew, M., Binger, R., Harmel, R., and Veith T.. Model evaluation guidelines for systematic quantification of accuracy in watershed simulations. T. ASABE, 50, 885–900, 2007.

Mutua, F. M.: The use of the Akaike Information Criterion in the identification of an optimum flood frequency model, Hydrolog. Sci. J., 39, 235–244, doi:10.1080/02626669409492740, 1994.

**Editor**

*The presentation of the BN performance in figs 5&8 is misleading as the values are given in m3/s, where results always look much nicer than if specific discharge values would be plotted (see figs 6&7 in Wrede et al. (2013) as example). Based on my comments&reply before publication in HESS-D, I understand the aim to compare results with previous papers (which used the m3/s comparison) and the drop in performance is now mentioned in one sentence. However, it would be much better if the plots would present the simulations in the fair way (i.e. specific discharge values) and the comparison would be given in words that vice versa. After all, we should aim at providing examples of good practices!*

We thank the editor for this observation. We added new graphs (Fig. 6 and 10) in order to present specific discharge better in the analysis, and added more related description. We noticed interestingly that the rank correlations for all four cases ($Q_{MAMX}$, $Q_{1000}$, $Q_{100}$, $Q_{10}$) when considering specific discharges are in the order of 0.8 and their bivariate distribution does not present large asymmetries (Fig. S5 in Supplement 2). This could be indication that a method based on copulas could also be used for bias correction however investigating this fact further falls out of the scope of this paper. This observation is also added to section 3.1.

*Opposite to the authors' interpretation of Rojas et al. (2012) different climate models (GCM/RCM combinations) can provide largely varying results. Therefore, the use of only one GCM/RCM combination is a clear limitation and largely ignoring the uncertainties caused by the climate models (see conclusions in Rojas et al. 2012; and recent papers by PhD students in my group (Teutschbein and Seibert, 2012; Addor et al., 2014). Furthermore, given the often significant biases in climate model simulations, the decision to not use any bias correction is surprising at least. Both decisions (only one model,*

*no bias correction) need to be motivated more convincingly. At least I would recommend the authors to test the effects of using another model and/or bias correction.*

We decided to make a new simulation with another EURO-CORDEX model run, with figures and comparison added to a new Supplement 2 file. We also included in that Supplement the result of running the BN model with ERA-Interim climate reanalysis – this model run was done before and mentioned in the paper, but we didn't present the results. The use of an alternative GCM-RCM combination (HadGEM2-ES-RACMO22E) gives similar results to the other model. We also amended the text to make more clear why the EC-EARTH-COSMO_4.8_clm17 was picked out of the various EURO-CORDEX model runs.

*My major concern, however, is figure 7, which illustrates predicted future changes in the 100-year floods over Europe. Such figures easily get the attention of media and decision makers and as scientists we, thus, really have to be careful in how we communicate such results. Presenting such results without also quantifying uncertainties is problematic. I, thus, strongly recommend to quantify the uncertainties of the results shown in fig. 7. There are partly huge differences in the predicted changes in neighboring streams (even in regions where snow is not important) and I am missing an*

*explanation of these differences. One approach to help understanding the patterns would be to use a uniform precipitation change, so that the effects of differences in catchment characteristics versus differences in precip-changes could be disentangled. (In addition, as far as I can see, the climate change prediction is not mentioned in the methods part.)*

We agree with the editor regarding this comment. We have made explicit throughout the paper that a full discussion of predictions of extreme discharge from our model under climate change is not the main scope of the paper but rather an example of application. Rather, our paper aims to show how the methodology can be applied to hydrologic research

We observe that large differences in neighbouring streams could be seen in any of the Lisflood publications we cite as well.. We added information to Fig. 7 and Table 4 which climate run was used to make the prediction. Also, we added information to section 2.5 about the use of climate predictions:

[revised manuscript text omitted]

---

## Author Response (AR2)

**Response to review**

We are thankful and pleased that more of our peers are willing to revise our work. We thank Dr. Anna Sikorska for her time and revision. Below we respond to her comments and indicate the changes made in the paper. The marked-up revised manuscript was added afterwards.

**General comments**

    **1.** *The presented method is tested with catchments of different areas ranging from less than 100 km2 (how much exactly?) to above 100'000 km2 (how much exactly?).*

The range of the catchments' size is mentioned in section 2.2: "The catchments' size spans from 1.4 to 807,000 km$^2$," However to make it
10 more explicit we include this information already in the abstract.

    *However, only daily discharges were used in the analysis which most likely is a limiting factor for small catchments (<100 km2), where peaks occur rather at an hourly scale. This is also confirmed by the validation results (Table 2.), where all efficiency metrics dropped for medium catchments (i.e. <500 km2) and most likely would drop even further for even smaller catchments*
15     *(i.e. <100 km2). Thus, I have concerns regarding the application of the method for such small catchments. Could you provide these estimates for different size groups as indicated in table 1 or explain the reason for aggregating all catchments into only two groups?*

It is true that the results are not as good for smaller catchments. We discuss in 4.2 that daily discharges are not well suited for small catchments: "More crucially, daily discharge is not adequate to model flash floods, or floods of short duration, in small catchments. These events can occur in matter of minutes, and do not even require a river bed".  We added the results for catchments of different sizes to table
20 2, as well as for specific discharge in that split, and modified the paragraph referring to it: "Detailed results in Table 2 show that the performance of the model drops for smallest catchments. Especially for those below 100 km$^2$  (177 catchments) the performance is poor. For others, above 500 km$^2$, the R$^2$ and $I_{NSE}$ values are mostly in the range of 0.5–0.6 for specific discharges, as when considering all stations."

    *Nevertheless, based only on the presented results, in my opinion, application of this methodology for catchments of a size*
25     *smaller than 500 km2 should be done with caution and for catchments smaller than 100 km2 should be evaluated prior to application. This issue should at least be stressed in the manuscript in both Discussion and Conclusions. Therefore, the conclusion saying that the method can be applied at any ungauged location (l. 78bp. 26) is not fully true (as it was not proven for small catchments) and thus should be corrected.*

- The reason why the data were split around 500 km$^2$ is because that value is close to the median size of catchments in the sample (650
30 km$^2$) and was used as a threshold in pan-European flood modelling by Alfieri et al. (2014).

- We agree with Dr. Sikorska that for small size catchments the applicability of the model may be further evaluated. Hence we have further modified the conclusions and discussion to make clear that we see lower performance for smaller catchments.

- We also clarify this fact in the sentence "The method can be applied to create basic flood scenarios at any ungauged location based on a few variables". What we meant was that the data necessary to apply the method can be found for any location using pan-European spatial
35 databases, not that the method is accurate in every location. Furthermore, we added a warning for potential users of the results (section 4.3) to be aware of the limited accuracy for very small catchments. This warning is repeated in the conclusion: "The accuracy at different ungauged locations however, as seen in Table 2, could differ". Finally, we may add that for reasons both of reducing computational time and inadequacy to model flash floods, only catchments >100 km$^2$ are used further in our follow-up study, which was published very recently as a discussion paper in *Natural Hazards and Earth System Sciences* (citation was added to the manuscript)

    **2.** *My second general issue concerns two important limitations of the presented method which are not mentioned in the manuscript (or without enough details): a) uncertainty in observed data of discharges used in this study, and b) assumption of stationarity of discharge data over observation period of 50 years. The first issue may be difficult to quantify but may have an impact on the analysis and results obtained and thus should be briefly mentioned. The second issue declines any climate impacts*
45     *or land use changes (briefly mentioned in the manuscript) over the analysed period of 50 years which may or may not be true for some of the analysed catchments. Both of these issues should at least be mentioned in the discussion or Introduction part.*

- We agree with Dr. Sikorska that different sources of uncertainty may occur. We have discussed the ones we consider most relevant in section 4.2 Apart from data there could also be model uncertainty for example. The main purpose of the paper however is to present a methodology that we show performs reasonably well in comparison to other methodologies used for the same purpose.  Errors in the
50 measurements might occur, however as we mention, "all datasets were quality-checked by the providers". We clarify now in this sentence that "only a few cases of misplaced decimals in daily series have been identified in the data after inspection". Current research carried out by our team points towards the fact that these errors contribute in weakening the probabilistic dependence of extreme discharge and other variables in the model. This in turn lowers the predictive accuracy of a given quantification. We have added this aspect to the discussion of uncertainties and limitations (4.2) and applications and future developments (4.3).

- Regarding stationarity, there are two aspects. There is no stationarity assumed in the BN, as it is quantified using annual maximum discharges ($Q_{AMAX}$) values from a 56-year period. Therefore, annual data on climate variables is also included. In fact, we mention in sections 2.3 and 4.2 about the problem that we only use land use data for one time point due to the lack of consistent data for the preceding periods, therefore weakening the relation between discharge and land use. Stationarity is, however, assumed in the extreme value analysis. Using Spearman's rank correlation we identified that in 918 of 1125 gauges used to obtain return period the trend is not significant at level of significance of 0.05. We added this information to section 2.6. Using only those filtered stations for validation had little (though positive) impact on the results.

**Major comments**

*1. P. 10, Section 2.2. and Table 1. The information about the number of stations available per country, although important, is not that much informative due to the different coverage of the stations. For instance, 90 stations in Switzerland cover a relatively higher area than 133 stations in Germany. Thus, more informative would be a relative value of the number of stations per country area or the density of stations in each country available for analysis. I would suggest to add this information to the table.*

The information on the number of stations per 1000 km$^2$ was added to the table.

*2. Figure 5 and 6 panels bd. The authors compare the simulated discharge peaks with return periods of 10year, 100year and 1000year to "observed" values. How these 1000yr observed values are obtained? To my understanding, these observed values are not really observed but are computed/extrapolated using 30 years of observed discharges. Please correct this notation properly.*

The axes of figure 5 and 6 were adjusted to "$Q_x$ based on simulated/observed data", as the return periods are based on observed or simulated annual maxima of discharges. The explanation that return periods were obtained by fitting $Q_{AMAX}$ to Gumbel distribution was already included in the figure's caption. For figures showing the mean of annual maxima, the change of axes description is not necessary, as the mean is directly calculated from observations/simulations. Fig. 6(c) was corrected as it accidently showed results for a 2-year return period instead of a 100-year period.

*3. Figure 7. I would expect that the current stage (i.e. observation years) is also included in this figure for a better comparison. In addition, I agree with the Editor that presenting this Figure without any uncertainty or critical consideration is a bit risky. Consider modifying the caption to make it clearer that these results are just simulations using specific emission scenarios.*

The caption in Fig. 7 (Fig. 8 after revision) was adjusted according to the reviewer's suggestion.

*4. P. 17 l. 2830. These sentences are slightly misleading. Are 30 years of observations used for calibration or validation or for both, and how do you split 50 years of observations into two subsets of 30 years? Providing the exact years, which were used for calibration, should solve this issue.*

The 30-year periods are used for validation, as mentioned in this paragraph, which we modified for clarity:

"The main validation set consists of 958 stations with 1971–2000 data, 129 with 1961–90 data and 38 with 1951–80 data; 1125 in total out of 1841 used to quantify the BN."

We also explain in section 2.2:

"Long data series, i.e. at least three full decades of uninterrupted data (1951–80, 1961–90 or 1971–2000) were available for 1125 stations. These observations were used to validate the accuracy of the model in estimating mean $Q_{AMAX}$ and return periods, while the complete database was used to quantify the Bayesian Network model."

*5. P. 18 l. 13. The case when simulations perfectly fit the observed data (also the line 1:1 when plotted) does not necessarily mean that no bias is present in simulations. Please note that the observed data usually contain measurement errors. Hence, if these errors are not represented, fitting the line 1:1 by simulations may in fact result from the model (simulation) bias. Hence, I suggest to change "no bias" into "perfect fit".*

We opted to use a "bias" measure, since the name is used in many of the studies cited in the paper and named as such, and also to avoid confusion with "fitting" the $Q_{AMAX}$ values to parametric distributions. Indeed the bias measure can be seen as fitting a 1:1 line, as we note in section 2.6. Therefore, we have rephrased sentences which mention "bias" in connection to $I_{NSE}$ indicator, so that "bias" is no longer mentioned in those cases.

*6. P. 20 l. 15 and table 2. What do you mean with the statement that the model performance is/remains acceptable? Could you provide quantitative values supporting this statement?*

We explain the threshold for 'acceptable' in the previous paragraph (a value of 0.5, according to Moriasi et al. 2007). The mention of 'acceptable' was also dropped from this paragraph after revising the paragraph in accordance to general comment no. 1.

*7. P. 20 l. 26 and afterwards and Fig. 7. The results that authors present do not demonstrate the real trends in extreme discharges in Europe but only the projected trends using specific emission scenarios. Please adopt the paragraph and figure caption respectively.*

The paragraph and caption in Fig. 7 (Fig. 8 after revision) were revised according to the reviewer's suggestion.

*8. P. 22. The entire paragraph presenting the estimation of RFA analysis fits better into the method and results section that into discussion.*

We have applied the reviewer's suggestion by moving the analysis involving RFA to sections 2.7 (methodology) and 3.1 (results). The text in other places was accordingly adjusted to accommodate this change.

> **9.** *P. 24 l. 8. Daily data are not adequate to model both flash floods and short duration floods (of rainfall duration shorter than a day). Both of these floods may occur in small catchments which in fact may explain poorer efficiency of the model in these catchments obtained by the authors.*

The paragraph was adjusted to explicitly mention short duration floods and small catchments.

> **10.** *P. 25 l. 7. The given number of 246 years of simulations is wrong. 56 years of observations plus two different emission scenarios each of 95 simulation years do not sum up to 246 years! Because emission scenarios cover the same period of time, only 151 years of simulations are analysed.*

As explained in the parentheses in that sentence, the number refers to simulation-years (56 + 2 X 95 = 246).We understand that that may be somewhat misleading, therefore we now mention 151 years with extra explanation.

> **11.** *P. 26 l. 13. Could you say something about the model performance when only some of variables are available at a given location when using BN? I think it would be worth checking (here or in the further work) how the model performance changes depending on the number of variables used/available and if this change is variable specific.*

We agree with the reviewer that this is a good point for the discussion and for future research. An example of the partially conditionalized BN can be found in the Supplement and Fig. 4. We have also added the following sentences in the discussion under future research:

"As a first check, Couasnon (2017) applied the model for the contiguous United States, indicating that the European quantification performed generally well, though much less accuracy was observed for arid and hurricane-influenced parts of the country than in those with temperate climate. Quantification based on US or combined (US/Europe) data performed less well, though for any variant the results were better than when using RFA, which was originally validated for that area by Smith et al. (2015). Finally, the model could be potentially evaluated not only using all variables, but conditionalized only on some of them, as all might not be available in a given location."

**Minor comments, typos and suggestions:**

> *There are also several typos which should be corrected. Below I present just some of them but please doublecheck your manuscript again.*
> **1.** *P. 16 l. 2930. Change into: "…i.e., the value of the node or set of nodes (other than discharges) is defined based on the new evidence…"*
> **2.** *P. 17 l. 14. Change "30 years of data" into "30 years of discharge observations"*
> **3.** *P. 17 l. 20. Change "other distributions" into "other tested distributions"*
> **4.** *P. 17 l. 27. There are more general sources for referring to the maximum likelihood estimation approach than the cited paper of Katz et al. 2012. E.g. Gelman et al. 2013 (Gelman, A., Carlin, J., Stern, H., et al. & Rubin, D. (2013).Bayesian data analysis (3rd ed.). London: Chapman & Hall/CR)*
> **5.** *P. 18 l. 6. Change into: "quality of return periods and average maxima simulations were…"*
> **6.** *P. 23 l. 19. Change "were" into "where"*
> **7.** *P. 23 l. 28. Change into "calculated as a fraction of …."*
> **8.** *P. 23 l. 32: change sentence into: "However, one alternative configuration worth mentioning is a BN incorporating terrain classification based on…"*
> **9.** *P. 24 l. 21. Change into "Climate data from CORDEX have/are of the…"*

We thank the reviewer for pointing all those issues, all changes have been applied to the manuscript and it was checked again for typos.

[revised manuscript text omitted]

---

## Author Response (AR4)

**Response to review**

We would like to thank Dr. Anna Sikorska for her time and review and prof. Seibert for his further recommendations. Below we respond to the comments and indicate the changes made in the paper. The marked-up revised manuscript was added

5   afterwards.

**The reviewer's comments**

**Comment (C):**

1. p.5. l.6: change into "Data on annual maxima of daily discharges from more than 1800 river gauge stations (with
10   catchment area ranging from 1.4 to 807,000 km2) were collected, together with information on terrain, land use and local catchments' climate."
2. p.9. l. 2: change "on the other hand" into "however"
3. p. 16 l. 19 remove "some" from supplementary data
4. p.18 l. 8: add comma before "especially" and remove "the performance is poor" from the sentence.
15   5. p.18 l. 34: add "even" before "better".
6. p.22 l. 16: remove "it"
7. p.22 l. 27: add "or floods" after floods of short duration, i.e.: "or floods in small catchments."
8. p.23 l. 1: "might still contain errors..."
9. p. 24 l. 12: change to "small and medium catchments..." if you use a threshold of 500km2.
20   10. p.25 l. 21: change into "increasing influence..."
11. Figure 8.: change "predicted" into "projected"

**Response (R):** All changes suggested by the reviewer we implemented as requested, except point 1, which was further modified according to our native speaker's language check:

*"Annual maxima of daily discharges from more than 1800 river gauges (stations with catchment areas ranging from*
25   *1.4 to 807,000 km$^2$) were collected, together with information on terrain, land use and local climate."*

**The editor's comments**

1. **C:** I have to say that the language partly could be improved and formulations could be more to the point. I list a few examples below. Please make sure to check your text once more very carefully to make sure formulations are as
30   clear as possible and to avoid language mistakes (perhaps seek advice from a native speaker or professional copyeditor).

**R:** We did a thorough re-read of the paper, making many corrections in the manuscript. We also employed the help of our colleague Antonia Sebastian from the USA to make a language check. She provided us with many corrections and suggestions for improvement, which we implemented in all sections.

2. **C:** Methods: "Some additional neighbouring basins were added for complete coverage of Europe except for the territory of the former Soviet Union. Only the outermost regions of Madeira, the Azores 15 and the Canary Islands were omitted because they are outside the EURO-CORDEX climate model; notwithstanding, their river networks are very limited and therefore of little interest."

No need to say that some rivers or of little interest (to which at least local people would disagree), just say that the islands in the Atlantic were excluded. These sentences also needs to be reformulated:

   a. Do you mean upstream parts of 'some' basins were added or entirely additional basins? The formulation with some.. added .. except … is a bit difficult to follow.

   b. I guess you excluded these islands entirely, but the sentence actually says only some regions on them were excluded, please clarify.

**R:** This paragraph was indeed unclear, therefore we rewrote it and now it reads the following:

*"Discharge data from measurement stations were collected over a domain covering most of Europe (Fig. 2). The study area includes the entire continent, plus Cyprus as a European Union (EU) member, with two exceptions. Out of the territory of the former Soviet Union, only river basins that are at least partially located within the EU were included. Also, the outlying regions of Madeira, the Azores and the Canary Islands were omitted because they are outside the EURO-CORDEX climate model's domain."*

3. **C:** Conclusions: "The accuracy at different ungauged locations however, as seen in Table 2, could differ." Please do not refer to a table here, but formulate the sentence (or two) so that it is clear what results you are referring to.

**R:** The sentence was reformulated as follows:

*"However, the accuracy at different ungauged locations varies to some degree. The best performance was found in Scandinavia, western Europe and the Danube basin, while the lowest was observed in southern Europe, especially in the Iberian Peninsula"*

4. **C:** "The advantages of our approach is that it is has low computational expense, it is explicit and flexible." Grammatical mistake :… are that. Furthermore, please be more specific, what do you mean by explicit and flexible in the context of this method.

**R:** The paragraph was rewritten so that the expressions 'explicit' and 'flexible' are placed together with their explanations:

[revised manuscript text omitted]